Report No.: **FERMILAB-PUB-25-0024-ETD**

First version submitted to arXiv on **February 11, 2025**
This version submitted to arXiv on **February 11, 2025**

# ARCANE Reweighting: A Monte Carlo Technique to Tackle the Negative Weights Problem in Collider Event Generation

**Prasanth Shyamsundar**[1⋆]

1 Fermi National Accelerator Laboratory, Batavia, Illinois 60510, USA

⋆ [prasanth@fnal.gov](mailto:prasanth@fnal.gov)

## Abstract

Negatively weighted events, which appear in the Monte Carlo (MC) simulation of particle collisions, significantly increases the computational resource requirements of current and future collider experiments. This paper introduces and theoretically discusses an MC technique called ARCANE reweighting for reducing or eliminating negatively weighted events. The technique works by redistributing (via an additive reweighting) the contributions of different pathways within an event generator that lead to the same final event. The technique is exact and does not introduce any biases in the distributions of physical observables. A companion paper demonstrates the technique for a physics example.

# 1  Introduction

The simulation of collider events under different theory models and model parameter values constitutes a significant part of the computational resource requirements of experiments at the Large Hadron Collider (LHC), as well as future colliders like the High-Luminosity LHC (HL-LHC) and the Electron–Ion Collider (EIC). Negatively weighted events, which feature, for instance, in the generation of particle-level events at beyond-leading-order-accuracy in perturbation theory [1, 2], significantly increase the number of simulated events needed by experiments to reach their target precisions in Monte Carlo (MC) predictions. This poses a significant computational challenge in collider physics [1–3]. This paper introduces a Monte Carlo technique, dubbed ARCANE reweighting[1], for reducing the negative weights problem in generic MC sampling tasks. As a specific application, ARCANE reweighting can be used in collider event generation, e.g, with the MC@NLO formalism [4–6]. In this use-case, ARCANE can reduce the fraction of negative weights by redistributing the contributions of "hard remainder" or $\mathbb{H}$-type events and "standard" or $\mathbb{S}$-type events

   a)  without introducing any biases in the distribution of physical observables,

   b)  without requiring any changes to the matching and merging prescriptions used, and

   c)  without introducing (i) any dependency between the sampled events or (ii) any uncertainties that will not be captured by the standard data analysis techniques used in HEP.

This is accomplished using an appropriately constructed additive correction to the event-weights. As will be seen later, ARCANE reweighting constitutes a non-trivial departure from the traditional MC generation procedures used in HEP. So, for pedagogical clarity, the technique is

---

[1]ARCANE stands for ARCANE Reweighting Can Avoid Negative Events.

discussed in this paper in a generic context, without bringing in the complexities of a specific HEP application like MC@NLO event generation. This paper uses standard probability and statistics notation, as opposed to the cross-section notation (featuring $d\sigma$-s) typically used in the HEP-MC literature, also for pedagogical clarity. An application of ARCANE reweighting to reduce the negative weights problem in the generation of $(e^+e^- \rightarrow q\bar{q} + 1\,\texttt{jet})$ events, using the MC@NLO formalism, is presented in a companion paper, Ref. [7]. This companion paper also discusses how ARCANE reweighting could be systematically applied to other collision processes of interest, including hadron collisions like $(pp \rightarrow t\bar{t} + \texttt{jets})$ and $(pp \rightarrow Z + \texttt{jets})$.

The rest of this paper is organized as follows. The rest of Section 1 introduces the negative weights problem and discusses some previous work on mitigating it, from a statistical perspective. Section 2 introduces the ARCANE reweighting technique for a generic MC generation task. Section 3 discusses the complementarity of ARCANE reweighting and rejection reweighting techniques, and how the two techniques can be used in tandem. Section 4 contains some quantitative discussions about the computational benefits of ARCANE reweighting. Section 5 discusses some generalizations of the basic ARCANE reweighting technique. Some concluding remarks are provided in Section 6. The problem of mutually dependency of events produced using some other reweighting strategies in the literature is discussed in Appendix A.

It may not be clear from the presentation in this paper how ARCANE reweighting could practically be implemented for collider physics scenarios. Also, many of the discussions in this paper are not necessary for following the demonstration in the companion paper, Ref. [7]. The description of the technique in Section 2 and some additional details in Section 2.2 and Section 2.3 are sufficient to follow Ref. [7]. These are also reviewed briefly in Ref. [7], for those who want to read it directly.

## 1.1 Some Notions and Notations

Some notions and notations are introduced here, in order to simplify the subsequent discussion of the negative weights problem and the ARCANE reweighting technique.

**Some common caveats.** Throughout this paper, any statistical property (like expectation value, variance, etc.) referenced in an equation is assumed to exist and be finite. Throughout this paper, two probability densities or two quasi-probability densities (defined below) will be considered equal if they only differ on a set of measure zero. Likewise, functions of a random variable $X$ (or statistical properties conditioned on the value of a random variable $X$) will be said to be equal if they are equal almost everywhere in the domain of $X$. For simplicity, the notation "$\forall$" will mean "for all" or "almost everywhere", depending on context.

**Probability densities.** Let the notation $P_X^{\langle\text{optional-superscript}\rangle}$ represent the probability density function[2], or probability density for short, of a generic data-attribute $X$ under some data-sampling procedure; the specific sampling procedure under consideration could be indicated by an optional superscript. The value of the probability density function at a given value of $X$, say $X = x$, is given by $P_X^{\langle\text{optional-superscript}\rangle}(x)$. The conditional probability density of an attribute $Y$ given $X$ will be indicated as $P_{(Y|X)}^{\langle\text{optional-superscript}\rangle}$; its value for $X = x$ and $Y = y$ will be written as $P_{(Y|X)}^{\langle\text{optional-superscript}\rangle}(y \mid x)$. Parameterized probability densities will be represented similarly, except with a semicolon in place of |.

Multiple probability density functions with the same superscript (including no superscript) correspond to densities under the same sampling procedure. In other words, $P_X$ and $P_Y$ should

---

[2]Probability and quasi-probability densities of the different data-attributes are defined using appropriate reference measures; such measures are assumed to exist for all data-attributes considered in this paper.

be thought of as marginal densities corresponding to the same joint density $P_{(X,Y)}$ of $(X, Y)$.

**Quasi-probability densities.**  This paper uses the notion of quasi-probability densities, or quasi densities for short, to facilitate the discussion of the negative weights problem. Quasi-probability densities, in the context of this paper, are similar to regular probability densities with two differences: a) they are not necessarily non-negative and b) they need not be unit-normalized.[3] Let the notation $F_X^{\langle \text{optional-superscript} \rangle}$ represent a quasi-probability density of $X$. $F_X$ can roughly be thought of as the differential cross-section of the attribute $X$.[4]

**Modeling quasi-densities with weighted densities.**  Weighted datasamples have a special, real-valued weight-attribute $W$, which a) serves to modify the distribution represented by the dataset, and b) is treated differently than other data-attributes, both during data generation and in subsequent data analyses. Consider a Monte Carlo procedure which produces weighted datasamples $(W, X) \in \mathbb{R} \times \mathcal{X}$ with the joint probability density $P_{(W,X)}^{\text{MC}}$, henceforth referred to as the *weighted density* of $X$. The weighted density $P_{(W,X)}^{\text{MC}}$ (or equivalently, the corresponding sampling procedure) will be said to model the quasi density $F_X$ if and only if the following condition is satisfied:

$$F_X(x) = P_X^{\text{MC}}(x)\, \mathbb{E}_{\text{MC}}\big[W \,\big|\, X = x \big] \quad \left( \equiv \int_{\mathbb{R}} \mathrm{d}w\; w\, P_{(W,X)}^{\text{MC}}(w, x) \right), \qquad \forall x \in \mathcal{X}, \qquad (1)$$

where $\mathbb{R}$ is the set of real numbers and $\mathbb{E}_{\text{MC}}[\cdot]$ represents the expectation under the sampling distribution $P^{\text{MC}}$. Note that here $W$ can be positive, negative, or zero. This definition of a sampling procedure modeling a given quasi density aligns with the typical goals of event generators and simulators in HEP.

  In the rest of this paper, "to sample weighted (possibly constant-weighted) events or datapoints from $F_X$" will mean to sample the datapoints from a $P_{(W,X)}^{\text{MC}}$ that models $F_X$. Due to the special nature of the event-weight attribute, throughout this paper, the notion of quasi density will be applicable only to event-attributes other than the event-weight, i.e., $F_W$, $F_{(W,X)}$, etc. will not be considered in this paper.

**Generic integrals.**  Integrations with respect to data-attributes will be assumed to be performed using appropriate reference measures, e.g., counting measure for discrete attributes. In this way summations over discrete attributes, integrals over continuous attributes, as well as complicated combinations of these can all be represented in a condensed form, as integrals. For example, if $X$ is a data-attribute that can either be of the form $(0, U)$ or of the form $(1, U, V)$, where $U$ and $V$ can be arbitrary real numbers, then integration with respect to $X$ over its domain $\mathcal{X} \equiv \big(\{0\} \times \mathbb{R}\big) \bigcup \big(\{1\} \times \mathbb{R}^2\big)$ is a shorthand for integration with respect to $U$ and $V$ as well as a sum over the first element:

$$\int_{\mathcal{X}} \mathrm{d}x\, f(x) \equiv \int_{\mathbb{R}} \mathrm{d}u\, f((0, u)) \;+\; \int_{\mathbb{R}} \mathrm{d}u \int_{\mathbb{R}} \mathrm{d}v\, f((1, u, v)). \qquad (2)$$

**Marginal quasi-probability densities.**  The joint quasi density $F_{(X,Y)}$ of the attributes $(X, Y)$ uniquely determines the marginal quasi densities $F_X$ and $F_Y$ of the attributes $X \in \mathcal{X}$ and $Y \in \mathcal{Y}$, respectively. As in the case of regular probability densities, integrating $F_{(X,Y)}(x, y)$

---

[3]This corresponds to relaxing the first two of Kolmogorov's three probability axioms.

[4]The only caveat here is that this paper will consider quasi densities for some non-standard data-attributes, which one typically would not associate differential cross-sections with.

with respect to $y$ over $\mathcal{Y}$ gives $F_X(x)$.[5] More generally, for any attribute $X$ that is fully determined by another attribute $Y$, $F_Y$ uniquely determines $F_X$. An important property, relevant for the following discussion, is that if $P^{\mathrm{MC}}_{(W,Y)}$ models $F_Y$, then for any attribute $X$ that is fully determined by $Y$, $P^{\mathrm{MC}}_{(W,X)}$ models $F_X$.

## 1.2  Latent Event Attributes in Monte Carlo Sampling

The pipeline used to produce detector-level simulated events in HEP consists of several sequential stages or steps. Let us split this pipeline into two parts at an arbitrary point.[6] To simplify the discussion, let this splitting point be within or immediately after the particle-level event generation phase.[7] Let $V$ be an attribute that captures all the information about an event (excluding its weight) produced by the first part of the pipeline that will be used by (or is "visible" to) the second part, as well as the subsequent data analysis pipeline. The purpose of the first part of the simulation pipeline can be thought of as producing weighted events from a quasi density $F^{\mathrm{PH}}_V$ that corresponds to some underlying physics theory model (with some specific values for the parameters of the model) within some desired level of accuracy, like leading order (LO), next-to-leading order (NLO), etc. in perturbation theory. For clarity, the superscript PH in $F^{\mathrm{PH}}_V$ can be interpreted as being short for "physics". In addition to $V$, the events produced by the first part of the pipeline could also have additional hidden or latent attributes that are not required for the subsequent processing of the event. Some of these hidden attributes will be necessitated by the theoretical/phenomenological/heuristic description of $F^{\mathrm{PH}}_V$. For example, let us say we write $F^{\mathrm{PH}}_V$ as

$$F^{\mathrm{PH\text{-}eg\text{-}1}}_V(v) \equiv F^{\mathrm{term\text{-}1}}_V(v) + F^{\mathrm{term\text{-}2}}_V(v), \tag{3}$$

and sample events from $F^{\mathrm{PH\text{-}eg\text{-}1}}_V$ by randomly sampling each event either from $F^{\mathrm{term\text{-}1}}_V$ or from $F^{\mathrm{term\text{-}2}}_V$.[8] Now, we have implicitly added a latent attribute to each event, which indicates which of the two terms the event was sampled from. This is the case, for example, in the MC@NLO formalism [4–6, 8, 9], where events could be of either the $\mathbb{S}$-type ("standard" events) or the $\mathbb{H}$-type ("hard remainder" events). The presence of a latent variable in this situation can be shown manifestly by writing $F^{\mathrm{PH\text{-}eg\text{-}1}}_V(v)$ as

$$F^{\mathrm{PH\text{-}eg\text{-}1}}_V(v) = \sum_{z=1}^{2} F^{\mathrm{PH\text{-}eg\text{-}1}}_{(V,Z)}(v,z), \qquad \text{where} \quad F^{\mathrm{PH\text{-}eg\text{-}1}}_{(V,Z)}(v,z) \equiv F^{\mathrm{term\text{-}z}}_V(v). \tag{4}$$

Here $Z$ is a latent data-attribute that gets marginalized over. Each $F^{\mathrm{term\text{-}z}}_V$ could likewise be written in terms of more hidden attributes, which are integrated or summed over. In addition to such theory/phenomenology-necessitated hidden variables, there could be additional hidden attributes originating from the implementation of the Monte Carlo procedure. For example, when generating a single parton shower emission using the veto algorithm [8–12], the specific sequence of emission scale proposals that are rejected before one is accepted (or the emission scale cutoff is reached) can be thought of as a hidden attribute of the event. This

---

[5]This follows from Kolmogorov's third probability axiom.

[6]Note that there could be multiple pathways within the simulation pipeline that an event could take. In this sense, the splitting "point" should be thought of as a criterion that will be satisfied at some point during the production of each event.

[7]Much of the discussion in this paper will hold even if the splitting point is within the detector simulation phase. However, the negative weights problem is typically encountered at the particle-level event generation stage, hence the simplifying assumption.

[8]It is assumed here that the event-weights will be appropriately scaled based on the probabilities of choosing each of the two terms.

particular hidden variable is not necessitated by theory; it is an artifact of the veto algorithm (or of the fact that the integral in the Sudokov form factor is difficult to perform analytically).

Let $H_{\mathrm{thr}} \in \mathcal{H}_{\mathrm{thr}}$ be a variable that represents all the hidden attributes of the event (produced before the chosen splitting point) that originate from the theoretical description of $F_V^{\mathrm{PH}}$. Likewise, let $H_{\mathrm{imp}} \in \mathcal{H}_{\mathrm{imp}}$ represent all hidden attributes of an event that originate from the implementation of the sampling procedure. Let $H \equiv (H_{\mathrm{thr}}, H_{\mathrm{imp}})$ represent the combination of both these types of latent information. The task of generating $V$ from $F_V^{\mathrm{PH}}$ can be broken down as follows.

**Step 1:** First, one writes $F_V^{\mathrm{PH}}$ in a form that is *"suitable for performing Monte Carlo sampling"* (this will be explained further in Section 2.1), using latent variables as follows:

$$ F_V^{\mathrm{PH}}(v) \equiv \int_{\mathcal{H}_{\mathrm{thr}}} \mathrm{d}h_{\mathrm{thr}} \; F_{(V,H_{\mathrm{thr}})}^{\mathrm{PH}}(v, h_{\mathrm{thr}}). \tag{5} $$

$F_{(V,H_{\mathrm{thr}})}^{\mathrm{PH}}$ can be thought of as the theory description for $F_V^{\mathrm{PH}}$. For particle-level event generation, this step involves several theoretical frameworks and techniques [8, 9, 13–15], including a) higher order matrix-element calculations, b) phenomenological models for showering, fragmentation, and hadronization, c) subtraction schemes, d) shower matching and merging prescriptions, e) parton distribution functions, etc. Note that integration with respect to $H_{\mathrm{thr}}$ in (5) could be a shorthand for a complicated aggregation over the different pathways in the MC pipeline that lead to events with the same value of $V$.

**Step 2:** Next, one creates an MC procedure for producing $(V, H_{\mathrm{thr}}, H_{\mathrm{imp}})$ with weighted density $P_{(W,V,H_{\mathrm{thr}},H_{\mathrm{imp}})}^{\mathrm{MC}}$ in such a way that $P_{(W,V,H_{\mathrm{thr}})}^{\mathrm{MC}}$ models $F_{(V,H_{\mathrm{thr}})}^{\mathrm{PH}}$. This sampling procedure can be used to sample weighted events distributed as per $F_V^{\mathrm{PH}}$. The integration in (5) is performed implicitly in this step, via the marginalization of the latent attributes. This step involves the use of Monte Carlo algorithms [14–16] like inverse transform sampling, importance sampling (including its various implementation-variants [17–23]), veto algorithm and its variants [8, 10, 11], and rejection reweighting. Software implementations of the event generation and simulation procedures is also a crucial aspect of this step [13–15, 24–28].

## 1.3 The Negative Weights Problem and Previous Work on Mitigating It

Even if $F_V^{\mathrm{PH}}$ is guaranteed to be non-negative, $F_{(V,H_{\mathrm{thr}})}^{\mathrm{PH}}$ in (5) could be negative for some values of $(V, H_{\mathrm{thr}})$. In such situations, in order to faithfully sample events from $F_{(V,H_{\mathrm{thr}})}^{\mathrm{PH}}$ one needs negatively weighted events or "negative events" for short. Such negative events are simply an artifact of the theoretical frameworks and techniques used to write $F_V^{\mathrm{PH}}$ in terms of $F_{(V,H_{\mathrm{thr}})}^{\mathrm{PH}}$. In the context of collider physics, negative events primarily arise in event generation at beyond-leading-order accuracy, e.g., using the MC@NLO formalism. The presence of negatively weighted events can significantly increase the amount of simulated data needed to reach desired precision levels in experimental analyses, and poses a major problem in terms of computational resources [1–3]. Next, let us review some previous work on mitigating this problem, using the simplified setup and notations of this paper. Uninterested readers should be able to skip this part and follow along from Section 2. Quantifying the negative weights problem is postponed until Section 4; all discussions in this paper regarding reducing the amount or degree of negative weights problem will be consistent with the quantification in Section 4.

A subtlety here is that, depending on the definition of $V$ (and the stages of the simulation pipeline under consideration), $F_V^{\mathrm{PH}}(v)$ can itself be negative for some values of $v$. This

is acceptable and *not necessarily* unphysical, since $V$ is not necessarily an experimentally observable event-attribute; it is only "visible" to the subsequent event generation and simulation stages. The techniques developed in this paper apply to such situations (where $F_V^{\text{PH}}$ is not non-negative) as well. However, for simplicity, such situations will mostly be ignored in the discussions and explanations in this paper until the nuanced quantitative discussions in Section 4.

### 1.3.1 Solutions From the Theory Side

One way to tackle the negative weights problem is to create a different theory description $F_{(V,H'_{\text{thr}})}^{\text{PH}'}$ (in "step 1" in Section 1.2) that has a reduced negative weights problem compared to $F_{(V,H_{\text{thr}})}^{\text{PH}}$ [29–33]. Here $H'_{\text{thr}}$ could possibly describe a different set of latent information than $H_{\text{thr}}$. For example, the POWHEG formalism [29, 30] is an alternative to MC@NLO for interfacing NLO matrix-element calculations and parton showers with a reduced negative weights problem. As another example, the MC@NLO-$\Delta$ formalism [32] is a careful modification of the standard MC@NLO to reduce the negative weights problem. Note that the different theory-formalisms may lead to slightly different quasi densities for $V$, i.e., $F_V^{\text{PH}'}$ may not exactly match $F_V^{\text{PH}}$. This is acceptable as long as $F_V^{\text{PH}'}$ also matches the underlying physics theory within the desired degree of accuracy; this leeway is actively exploited, e.g., in the MC@NLO-$\Delta$ approach.

The main difficulty of such approaches from the theory side is that such alternative theoretical formalisms or descriptions are not readily available for many important scenarios or processes of interest in HEP [1, 2]. Creating a formalism that interfaces higher order matrix-element calculations with parton showers in a consistent manner is already a challenging task; requiring the formalism to also have good statistical properties only makes this task harder. Furthermore, in practice, depending on the collision process under consideration, there could be a significant fraction of negative events even under such an alternative formalism (when one is available).

### 1.3.2 Positive Resampling and Related Techniques

An approach to tackle the negative weights problem from the statistics side is by using the so-called "positive resampling" and other related techniques [34–38]. Starting from a dataset of weighted datapoints, say $D = \left\{ (W_i, V_i) \right\}_{i=1}^{N}$, produced using existing generators, positive resamplers produce a modified dataset $D' = \left\{ (W'_i, V'_i) \right\}_{i=1}^{N'}$, with fewer or no negatively weighted events. Note that $D'$ is created simply by processing $D$, so these techniques will be subject to certain statistical limitations originating, e.g., from the data processing inequality [39].

One of the guiding principles behind positive resamplers is that events with the same value of $V$ are practically identical, for the purposes of the subsequent steps of the simulation pipeline as well as the subsequent data analysis pipelines. So, these techniques locally redistribute the weights of events that are close to each other[9] in the phase space of $V$. This can be done either by explicitly grouping events by proximity and redistributing their weights as in Ref. [34, 36–38], or implicitly with a neural network based reweighter trained on the original dataset, as in Ref. [35]. In this way, the contributions of positive events can be made to cancel out the negative weights, leading to an ostensible reduction in the negative weights problem. In this redistribution approach, only the event weights are changed and the visible event-attributes $V'_i$-s are simply copied over from the original dataset, hence the name "resampling". These techniques could also involve an optional [34, 36] or a mandatory [35] downsampling via rejection reweighting, so $N'$ could be less than $N$. Since the contributions are being smeared over a non-zero volume in the phase space of $V$, such techniques could introduce a bias in

---

[9]Different techniques, explicitly or implicitly, use different notions of distance to capture this closeness.

the quasi density of $V$. It was argued in Ref. [36] that this bias could be sufficiently small in a given situation, depending on the size of the original dataset that is being resampled, the phase space resolution that can be discerned or probed by a given experiment, the level of accuracy needed, etc.

A major, and arguably underappreciated, issue with such positive resampling techniques is that they produce dependent datapoints; the different datapoints influence each other's weights, so $(W_i', V_i')$ cannot be assumed to be independent of $(W_j', V_j')$ for $i \neq j$. Consequently, error estimation formulas derived assuming the mutual independence of events will not work for analyses performed using such resampled datasets. This is explained further in Appendix A.

Closely related to the positive resampling techniques is another class of techniques [40,41], which operate as follows: First a machine learning (ML) model is trained on a weighted training dataset produced using an existing event generator. Subsequently, the ML model is used to produce datapoints (one at a time) with a similar quasi density as the training dataset, but with fewer negative weights, or more generally with a reduced weight-variance. Alternatively, the ML model may also be used to reweight new, independent events, one at a time.[10] The events generated using such procedures are mutually independent of each other, **if** one treats the trained ML model as a fixed object (i.e., the events are independent, conditional on the trained ML model). However, the quasi distribution $F_V^{\mathrm{ML}}$ corresponding to the trained ML model will not exactly match the $F_V^{\mathrm{PH}}$ one is trying to sample from. If the trained ML model needs to be treated as a fixed object, then this discrepancy will have to be estimated/upperbounded and shown to be small enough for the purposes of an analysis, in order to use such ML-produced datasets. Alternatively, one could treat the trained ML model itself as a random object, say $\Lambda$, in order to estimate the ML-related uncertainties, as in Ref. [42]. However, now the ML-produced datasamples cannot be treated as mutually independent, due to their dependence on the common (random) $\Lambda$. Succinctly, if $\{U_i\}_{i=1}^N$ represents the (possible weighted) dataset produced by such procedures, then

$$P_{(U_1,\ldots,U_N|\Lambda)}(u_1,\ldots,u_N \mid \lambda) \;=\; \prod_{i=1}^N P_{(U_1|\Lambda)}(u_i \mid \lambda) \qquad \begin{pmatrix}\text{illustrates conditional}\\ \text{mutual independence}\end{pmatrix}, \quad (6)$$

$$P_{(U_1,\ldots,U_N)}(u_1,\ldots,u_N) \;\neq\; \prod_{i=1}^N P_{U_1}(u_i) \qquad \begin{pmatrix}\text{illustrates dependence}\\ \text{between datapoints}\end{pmatrix}. \quad (7)$$

The discussion in Appendix A, regarding the mutual dependency of datapoints, largely applies to such ML-based approaches as well.

### 1.3.3 Solutions From the Monte Carlo Side

Finally, let us review some approaches for tackling the negative weights problem from the Monte Carlo side. In general, these techniques operate by reorganizing or redistributing the contributions of the different pathways in the simulation pipeline that lead to the same visible event $V$, in such a way that the quasi density of $V$ remains unchanged.

**Folded integration and related methods.** Consider a quasi density $F_V^{\mathrm{PH\text{-}eg\text{-}2}}$ written as[11]

$$F_V^{\mathrm{PH\text{-}eg\text{-}2}}(v) = \int_{\mathcal{Z}} dz\, f(v,z), \qquad \text{with } \mathcal{Z} \equiv [0,1], \quad (8)$$

---

[10]This is slightly different from the resampling approach in [35]; here one dataset is processed in order to reweight another dataset, instead of itself.

[11]Throughout this paper, lower case $f$-s (possibly with subscripts, superscripts, and/or accents) are used to represent quasi densities that can be analytically or numerically computed.

where $z$ is a latent variable and $f$ is a function that a) can be analytically or numerically computed for different input values and b) can be positive, negative, or zero. One can sample events from $F_V^{\text{PH-eg-2}}$ directly by sampling $(V, Z)$ as per $f$ (using importance sampling), albeit with some negatively weighted events. The associated negative weights problem can be mitigated by first rewriting $F_V^{\text{PH-eg-2}}$ as

$$F_V^{\text{PH-eg-2}}(v) = \sum_{k=1}^{K} \int_{(k-1)/K}^{k/K} dz\, f(v, z) = \int_0^1 dt\, \widetilde{f}(v, t), \tag{9}$$

$$\text{where} \qquad \widetilde{f}(v, t) \equiv \frac{1}{K} \sum_{k=1}^{K} f\left(v\, ,\, \frac{k-1+t}{K}\right), \tag{10}$$

for some integer $K$. Now, one can sample events from $F_V^{\text{PH-eg-2}}$ by importance sampling $(V, T)$ from $\widetilde{f}$. The summation in (9), which is performed explicitly numerically (instead of implicitly, via MC), allows for some positive and negative contributions to possibly cancel out, thereby reducing the negative weights problem. This comes with an increase in the particle-level computational cost per datapoint sampled, since the computation of $\widetilde{f}(v, t)$ involves $K$ computations of the function $f$. This is typically more than offset by the computational benefits, in the detector simulation stage, from reducing the negative weights problem. This technique can easily be extended in a number of ways, e.g., a) to allow for an unequal splitting of the integration domain, b) to handle multidimensional integrals over $\mathcal{Z} \equiv [0, 1]^N$ (or a possibly more complicated domain), etc. This technique is referred to as the "folding approach" [43], since the integration region has been "folded over" $K$ times. The folding approach works by splitting the integration domain, say $\mathcal{Z}$, into $K$ mutually exclusive and collectively exhaustive regions with bijective maps between them, and stacking or folding them over each other. Even in situations where a convenient splitting of this kind cannot be found, one can still reduce the negative weights problem by combining the contributions of different $Z$-values, e.g., as follows

$$F_V^{\text{PH-eg-2}}(v) = \int_{\mathcal{Z}} dz_1 \cdots \int_{\mathcal{Z}} dz_K \underbrace{P_{(V,Z_1,\ldots,Z_K)}^{\text{MC}}(v, z_1, \ldots, z_K)}_{\text{sampling distribution}} \underbrace{\frac{\left[\int_{\mathcal{Z}} dz\right]^{(1-K)} \frac{1}{K} \sum_{k=1}^{K} f(v, z_k)}{P_{(V,Z_1,\ldots,Z_K)}^{\text{MC}}(v, z_1, \ldots, z_K)}}_{\text{weight}} \tag{11a}$$

$$F_V^{\text{PH-eg-2}}(v) = \int_{\mathcal{Z}} dz_1 \cdots \int_{\mathcal{Z}} dz_K \underbrace{P_{(V,Z_1,\ldots,Z_K)}^{\text{MC}}(v, z_1, \ldots, z_K)}_{\text{sampling distribution}} \underbrace{\left[\frac{1}{K} \sum_{k=1}^{K} \frac{f(v, z_k)}{P_{(V,Z_k)}^{\text{MC}}(v, z_k)}\right]}_{\text{weight}} \tag{11b}$$

$$F_V^{\text{PH-eg-2}}(v) = \int_{\mathcal{Z}} dz_1 \cdots \int_{\mathcal{Z}} dz_K \underbrace{\left[P_V^{\text{MC}}(v) \prod_{k=1}^{K} P_{(Z|V)}^{\text{MC}}(z_k \,|\, v)\right]}_{\text{sampling distribution}} \underbrace{\left[\frac{1}{K} \sum_{k=1}^{K} \frac{f(v, z_k)}{P_{(V,Z)}^{\text{MC}}(v, z_k)}\right]}_{\text{weight}} \tag{11c}$$

Each of these equations represents a way to sample and weight datapoints of the form $(V, Z_1, \ldots, Z_K)$, in a way that reduces the negative weights problem while preserving the quasi density of $V$.[12]

**Born spreading.** Consider a quasi density $F_V^{\text{PH-eg-3}}$ written as

$$F_V^{\text{PH-eg-3}}(v) = f_1(v) + \int_{\mathcal{Z}} dz\, f_2(v, z), \tag{12}$$

---

[12]There are certain conditions on $P^{\text{MC}}$ in each case in (11) to ensure proper coverage and finiteness of weights.

where $f_1$ and $f_2$ can be computed analytically or numerically for different input values and $f_2$ can be positive, negative, or zero. One can sample events from $F_V^{\text{PH-eg-3}}$ by probabilistically sampling some events from the term $f_1$ and others from the term $f_2$. Alternatively, one can write $F_V^{\text{PH-eg-3}}$ as

$$F_V^{\text{PH-eg-3}}(v) = \int_{\mathcal{Z}} \mathrm{d}z \; \overline{\overline{\tilde{f}}}(v, z),\tag{13a}$$

$$\text{where} \qquad \overline{\overline{\tilde{f}}}(v, z) \equiv f_1(v)\, f_{\text{spread}}^{\text{unit-norm}}(z) + f_2(v, z),\tag{13b}$$

and importance sample $(V, Z)$ as per $\overline{\overline{\tilde{f}}}$. Here $f_{\text{spread}}^{\text{unit-norm}}$ is an arbitrary unit-normalized quasi density of $Z$. A common approach here is to choose $f_{\text{spread}}^{\text{unit-norm}}(z)$ to be the uniform distribution over $\mathcal{Z}$. However, as pointed out in Ref. [44], the flexibility in $f_{\text{spread}}^{\text{unit-norm}}$ can be used to suitably *spread* the contribution of $f_1(v)$ over the different values of $Z$, in order to effectively cancel out some negative contributions from $f_2(v, z)$; this is referred to as "Born spreading" in Ref. [44], because in the context of that paper, $f_1$ represented the Born contribution to $\mathbb{S}$-type events in MC@NLO. Ref. [44] also provides a specific choice for $f_{\text{spread}}^{\text{unit-norm}}$ that is non-negative, but this restriction is not necessary for the validity of the technique.

**Parametric Control Variates.** The quasi density $F^{\text{PH-eg-2}}$ in (8) can be rewritten as

$$F_V^{\text{PH-eg-2}}(v) \equiv \int_{\mathcal{Z}} \mathrm{d}z \; \overline{\overline{\tilde{f}}}(v, z) \equiv \int_{\mathcal{Z}} \mathrm{d}z \left[ f(v, z) + g(v, z) \right],\tag{14}$$

where $g$ is a real-valued function that satisfies $\int_{\mathcal{Z}} \mathrm{d}z \; g(v, z) = 0$, for all $v \in \mathcal{V}$. Now one can sample events from $F_V^{\text{PH-eg-2}}$ by importance sampling $(V, Z)$ as per $\overline{\overline{\tilde{f}}}$. The flexibility in choosing $g$ can be used to reduce the negative weights problem in the original approach of sampling directly from $f$.[13] This is referred to in the literature as the parametric control variates technique [45]. This is similar to the control variates variance reduction technique [46–48] (used in MC integration), where a control $g(z)$, whose integral is known to be 0,[14] is added to the integrand of interest $f(z)$; the added intricacy here is that both $f$ and $g$ have an additional parameter $v$.

**Discussion.** For simplicity, let us refer to the techniques considered in Section 1.3.3 as *Monte Carlo approaches or techniques*. Like the solutions from theory side, the Monte Carlo approaches also work by creating an alternative quasi density with a reduced negative weights problem. The somewhat arbitrary distinction made in this paper between these two classes of techniques is as follows. While the theory solutions require a solid understanding of the underlying physics, the Monte Carlo solutions rely only on a) understanding the mechanics of an existing event generation pipeline, and b) knowing which event-attributes are important or visible to the subsequent stages and which attributes are (or can be made to be) auxiliary or unimportant. In this sense, as will be seen later, the ARCANE reweighting technique of this paper is also purely a Monte Carlo technique.

Similar to the positive resampling techniques, the Monte Carlo techniques also perform a form of redistribution or reorganization, but with some important differences. While the

---

[13]This technique can also be applied to $F_V^{\text{PH-eg-3}}$ in (13), with the original $f_{\text{spread}}^{\text{unit-norm}}$ set to the uniform distribution. There is a one-to-one correspondence between the improved $f_{\text{spread}}^{\text{unit-norm}}$ function used in the Born spreading technique and the function $g$ in the parametric control variates technique.

[14]In the standard presentation of the control variates technique, the control need not integrate to 0 and only needs to have a known integral. Such a control can be converted into one that integrates to 0, using a $z$-independent additive shift.

redistribution happens at the *sample* level (i.e., over a finite dataset) in the resampling techniques, it happens at the *population* level (in the statistics theory sense of the word [49]) in the Monte Carlo approaches. Furthermore, the reorganization in the Monte Carlo approaches is performed only among events with the exact same value of $V$, so there are strictly no biases introduced in the distributions of physical observables. The events can also be treated as being IID without any issues,[15] unlike in the case of resampling techniques.

Note that the techniques in Section 1.3.3 all operate, roughly speaking, within a single step in the event generation pipeline. For example, folded integration modifies a single importance sampling step that samples from $f(v,z)$ into one that samples from $\widetilde{f}(v,t)$, while the parametric control variates technique modifies $f(v,z)$ to $\widetilde{\widetilde{f}}(v,z)$. A limitation here is that these techniques cannot be used in a straightforward manner to perform a redistribution of contributions across more complicated event generation pathways encountered, e.g., in the MC@NLO formalism. The ARCANE reweighting technique offers a way to lift this limitation, while remaining similar in spirit to the Monte Carlo approaches discussed so far.

## 2   ARCANE Reweighting

In this paper, ARCANE reweighting technique will be presented simply as a way to modify an existing event generation pipeline, without making any reference to the underlying physics. Classifying the hidden attributes into theory-necessitated ones ($H_{\mathrm{thr}}$) and ones arising from implementation ($H_{\mathrm{imp}}$) was useful for discussing the negative weights problem in Section 1, but this distinction will be ignored presently. Likewise, the physics theory description $F^{\mathrm{PH}}_{(V,H_{\mathrm{thr}})}$ for the quasi density of $V$, on which the original event generation pipeline was based, will not be referenced.

Let the weighted distribution of $(V,H)$ under a given Monte Carlo sampling procedure be $P^{\mathrm{MC}}_{(W,V,H)}$. As a reminder, $V$ represents all the event-attributes that will be used by the subsequent stages of the simulation pipeline, and $H$ represents all the hidden event-attributes produced in the sampling procedure. Together, let $V$ and $H$ cover all sources of randomness in the event generation pipeline. This implies that the weight under the given Monte carlo procedure is uniquely determined by $(V,H)$, and can be written as

$$W_{\mathrm{MC}} \equiv \frac{F^{\mathrm{MC}}_{(V,H)}(V,H)}{P^{\mathrm{MC}}_{(V,H)}(V,H)}, \tag{15}$$

where $F^{\mathrm{MC}}_{(V,H)}$ is the quasi density of $(V,H)$ modeled by $P^{\mathrm{MC}}_{(W,V,H)}$. The subscript MC has been introduced to the event-weight, anticipating the reweighting to be introduced in this section. Analogous to (14), a simple way to modify the quasi density of $(V,H)$ without affecting the quasi density of $V$ is as follows:

$$F^{\mathrm{ARCANE}}_{(V,H)}(v,h) = F^{\mathrm{MC}}_{(V,H)}(v,h) + G_{(V,H)}(v,h), \tag{16}$$

where $G_{(V,H)}$ is a function that integrates over $H$ to zero, for every possible value of $V$. Under this construction, we have $F^{\mathrm{ARCANE}}_V \equiv F^{\mathrm{MC}}_V$, even though $F^{\mathrm{ARCANE}}_{(V,H)} \not\equiv F^{\mathrm{MC}}_{(V,H)}$. The flexibility in choosing $G$ allows one to redistribute contributions so that $F^{\mathrm{ARCANE}}_{(V,H)}$ can have a reduced (possibly zero) negative weights problem when compared to $F^{\mathrm{MC}}_{(V,H)}$. Henceforth, $G_{(V,H)}$ will be referred to as the "ARCANE redistribution function".

---

[15]This is true even if there are sources of randomness in the implementation of the Monte Carlo techniques, e.g., in choosing the function $f^{\mathrm{unit\text{-}norm}}_{\mathrm{spread}}$ in (13).

A key question here is, assuming one has access to an appropriate $G_{(V,H)}$, how does one construct an event sampling procedure that models $F^{\text{ARCANE}}_{(V,H)}$?[16] One idea is to randomly sample some events from $F^{\text{MC}}_{(V,H)}$ and others from $G_{(V,H)}$. However, this would add one more latent variable to the setup, which indicates which of the two terms a given event was sampled from. As such, this approach cannot reduce the negative weights problem in the event generation procedure; the events sampled from $F^{\text{MC}}_{(V,H)}$ will continue to have the same fraction of negative weights and the non-zero-weighted events sampled from $G_{(V,H)}$ will be 50% negative (after unweighting), by construction.

On the other hand, in order to model the quasi density $F^{\text{ARCANE}}_{(V,H)}$ without introducing any additional latent variables, the sampling procedure needs to include the contributions from **both** $F^{\text{MC}}_{(V,H)}$ and $G_{(V,H)}$ in **each** sampled event $(V,H)$. This way some positive and negative contributions can get cancelled at the single-event-weight level, leading to an overall reduction in the negative weights problem. The proposal in this paper is to accomplish this via an additive reweighting; a weighted event $(W_{\text{MC}}, V, H)$ produced by the original sampling procedure will be reweighted to $W_{\text{ARCANE}}$, given by

$$W_{\text{ARCANE}} = W_{\text{MC}} + W_\Delta, \qquad \text{where} \quad W_\Delta \equiv \frac{G_{(V,H)}(V,H)}{P^{\text{MC}}_{(V,H)}(V,H)}. \tag{17}$$

It is easy to see that the resulting modified sampling procedure will model $F^{\text{ARCANE}}_{(V,H)}$, as per the definition in (1). This is the core idea of ARCANE reweighting. Note that this technique is not just useful for reducing the negative weights problem. More generally, it can be used to reduce the weight-variance among events with the same value of $V$ but different values of $H$.

In order to compute the additive reweighting term $W_\Delta$, one should a) have access to the value of $(V,H)$ for each event and b) be able to compute $P^{\text{MC}}_{(V,H)}(v,h)$ for different input values $(v,h)$; this is discussed further in Section 2.4. Furthermore, one needs to construct a suitable redistribution function $G_{(V,H)}$; the implementation of this step will depend on the particular use-case at hand and will not be considered in detail in this paper. This paper only contains a high-level discussion of the optimal redistribution function in Section 2.3 and a few pointers on how to construct $G_{(V,H)}$ in Section 2.4. The companion paper, Ref. [7], contains a construction of $G_{(V,H)}$ for MC@NLO ($\text{e}^+\text{e}^- \longrightarrow \text{q}\bar{\text{q}} + 1\,\text{jet}$) events.

A useful analogy here is between ARCANE reweighting technique and the importance sampling algorithm. The high-level importance sampling algorithm only provides the framework for sampling from a target quasi density by a) sampling datapoints from an auxiliary probability density and b) subsequently weighting the datapoints by the ratio of the target and auxiliary densities. The actual implementation of importance sampling involves constructing an appropriate auxiliary sampling distribution for the specific situation at hand, e.g., using the VEGAS algorithm [17].

## 2.1 What is New Here: Escaping the "Forward Chain Monte Carlo" Paradigm

The preceding discussion on (abstaining from) implementing the redistribution function may feel unsatisfactory to the reader. ARCANE reweighting only seems to have replaced the original, difficult theory problem of constructing an alternative theory description $F^{\text{MC}'}_{(V,H_{\text{thr}})}$, as discussed in Section 1.3.1, with a new problem of constructing a redistribution function $G_{(V,H)}$. It may not be clear a priori why this new problem would be easier than the old one. However, despite the seeming superficiality of the technique, ARCANE reweighting changes the existing HEP event generation framework in a fundamental way, as discussed next.

---

[16]Unlike in (14), $F^{\text{MC}}_{(V,H)}$ and $G_{(V,H)}$ may be functions of complicated event-attributes $(V,H)$. It may not be feasible to efficiently sample and weight the full event information $(V,H)$ in a single step, say using importance sampling.

**A Chain Rule in Simulations and the Forward Monte Carlo Paradigm.** Let us say we are interested in sampling a random datapoint $(A, B) \in \mathcal{A} \times \mathcal{B}$ from a quasi density $F_{(A,B)}^{\mathrm{PH}}$ which is expressed as follows:

$$F_{(A,B)}^{\mathrm{PH}}(a, b) \equiv F_A^{\mathrm{PH\text{-}stage\text{-}1}}(a) \times F_{(B;A)}^{\mathrm{PH\text{-}stage\text{-}2}}(b\,;a)\,. \tag{18}$$

Furthermore, let us say we have access to weighted datasampling procedures that separately model $F_A^{\mathrm{PH\text{-}stage\text{-}1}}$ and $F_{(B;A)}^{\mathrm{PH\text{-}stage\text{-}2}}$, respectively. The first datasampling procedure produces a weighted datapoint $(W_1, A)$ with probability density $P_{(W_1,A)}^{\mathrm{MC\text{-}stage\text{-}1}}$. The second datasampling procedure takes the value of $A$ as a parameter and produces $(W_2, B)$ with the parameterized probability density $P_{(W_2,B;A)}^{\mathrm{MC\text{-}stage\text{-}2}}$. For concreteness, the relevant quasi densities and weighted densities are related as follows

$$F_A^{\mathrm{PH\text{-}stage\text{-}1}}(a) = \int_{\mathbb{R}} \mathrm{d}w_1\; w_1\; P_{(W_1,A)}^{\mathrm{MC\text{-}stage\text{-}2}}(w_1, a)\,, \qquad \forall a \in A, \tag{19}$$

$$F_{(B;A)}^{\mathrm{PH\text{-}stage\text{-}2}}(b\,;a) = \int_{\mathbb{R}} \mathrm{d}w_2\; w_2\; P_{(W_2,B;A)}^{\mathrm{MC\text{-}stage\text{-}2}}(w_2, b\,;a)\,, \qquad \forall (a, b) \in \mathcal{A} \times \mathcal{B}. \tag{20}$$

Importantly, the sampling procedure for the second stage does not depend on the value of $W_1$. Now, one can sample a datapoint from the quasi density $F_{(A,B)}^{\mathrm{PH}}$ as follows: i) First one samples $(W_1, A)$ using the first sampling procedure. ii) Next one samples $(W_2, B)$ using the second sampling procedure, using the value of $A$ from the first stage. iii) Finally one computes the weight of the datapoint as $W = W_1 W_2$ and returns $(W, A, B)$ as the sampled datapoint. It can be easily verified that this procedure does indeed model $F_{(A,B)}^{\mathrm{PH}}$ as per the definition in (1). Let us call this principle the "chain rule" in simulations.

This chain rule is ubiquitous in collider event generation and simulation. For example, it is unconsciously used in the assumption that if a weighted sampling procedure correctly models a given quasi density of some event-attribute, say $A$, it also models the corresponding quasi densities of all deterministic and non-deterministic transformations of $A$.[17] As another example, the fact that the event weights evolve multiplicatively through a forward Monte Carlo procedure is used when one reweights a simulated dataset to make it correspond to, say, a different value of a simulation parameter, e.g., as in matrix-element, parton shower, and/or hadronization reweighting [50–56].

More pertinently, the chain rule is used repeatedly to build complex models for the production of a collider event, using simpler models for smaller steps. Note that the random sampling step to be performed at some point in the generation pipeline could itself depend on the outcomes of the previous steps. For instance, an MC@NLO event evolves in completely different ways depending on whether it was chosen to be of the $\mathbb{H}$- or $\mathbb{S}$-type. Such situations can be emulated in the two-stage setup above as follows: Let the domain $\mathcal{B}$ of the attribute $B$ be the union of two different sets, $\mathcal{B}_{\mathrm{case\text{-}1}}$ and $\mathcal{B}_{\mathrm{case\text{-}2}}$. Let $F_{(B;A)}^{\mathrm{PH\text{-}stage\text{-}2}}$ be appropriately piece-wise defined so that it has support only in $\mathcal{B}_{\mathrm{case\text{-}1}}$ for some values of $A$, and only in $\mathcal{B}_{\mathrm{case\text{-}2}}$ for other values of $A$. Now, depending on the value of $A$, one would either sample $B$ from $\mathcal{B}_{\mathrm{case\text{-}1}}$ or from $\mathcal{B}_{\mathrm{case\text{-}2}}$.

Let any sampling procedure created using repeated application of the chain rule be referred to as a "Forward Chain Monte Carlo" (FCMC) procedure. The smallest steps of an FCMC procedure are typically implemented in HEP using one of a) inverse transform sampling, b) importance sampling, c) (possibly weighted) rejection-sampling, d) the veto algorithm[18], e)

---

[17]This follows by setting $F_{(B;A)}^{\mathrm{PH\text{-}stage\text{-}2}}$ to the transfer function corresponding to the transformation.

[18]The veto algorithm itself can be thought of as replacing a single showering step with a telescoping chain of proposal and accept-reject steps, which can be performed using inverse transform sampling and (possibly weighted) rejection-sampling, respectively.

random sampling from a discrete set of choices, etc., possibly conditioned on the outcomes of the previous steps.[19] In Section 1.2, a vague notion of writing $F_V^{\mathrm{PH}}$ "in a form that is suitable for performing Monte Carlo sampling" was used. In more concrete terms, this means writing down a quasi density $F_{(V,H_{\mathrm{thr}})}^{\mathrm{PH}}$ in a form that is suitable for performing FCMC sampling with, i.e., as a chain constructed from analytically or numerically relatively-easy-to-compute functions. These functions influence the different FCMC steps that sequentially produce, in some order, the various event-attributes encapsulated by the meta-attributes $V$ and $H_{\mathrm{thr}}$. The previous "searches" for theory formalisms or descriptions with a reduced sign problem were performed within this restricted, admittedly vaguely defined, class of "straightforwardly FCMC-able" formalisms. Even the Monte Carlo approaches like folded integration and Born spreading, discussed in Section 1.3.3, only modify relatively isolated steps within the generation pipeline, and otherwise operate within the realm of FCMC.

Like the original quasi density $F_{(V,H)}^{\mathrm{MC}}$, the redistribution function $G_{(V,H)}$ may itself be written as a chain of simpler functions (as discussed in Section 2.4.2). Furthermore, the event-attributes may appear in different orders in the chained expressions for $G_{(V,H)}$ and $F_{(V,H)}^{\mathrm{MC}}$. Importantly, the expression for $F_{(V,H)}^{\mathrm{ARCANE}}$ in (16) need not be FCMC-able *in a straightforward manner*[20] for ARCANE reweighting to be viable. In this sense, when exploring different options for the redistribution function $G_{(V,H)}$ under the ARCANE technique, one is exploring a space of solutions to the negative weights problem that was not as easily accessible under previous "searches" for theory formalisms with a reduced sign problem.

The success of the ARCANE reweighting technique relies crucially on whether or not a good redistribution function can be constructed for a given use-case. The hypothesis here is that constructing good reweighting functions will be relatively straightforward for many situations of interest in collider event generation. This stems from the intuition that it should be easier to construct a suitable function $G_{(V,H)}(v,h)$ that integrates over $h$ to **zero**, than it would be to construct an alternative, **easily FCMC-able**, function $F_{(V,H'_{\mathrm{thr}})}^{\mathrm{PH}'}(v,h'_{\mathrm{thr}})$ that integrates over $h'_{\mathrm{thr}}$ to (at least sufficiently approximately) equal the typically **intractable** quantity $F_V^{\mathrm{PH}}(v)$. The demonstration in the companion paper, Ref. [7], is intended to convince the community that ARCANE reweighting offers a practicable solution to the negative weights problem in HEP event generation.

## 2.2 Requirements on the Redistribution Function

As with importance sampling, we need certain conditions to ensure "proper coverage" and finiteness of weights in ARCANE reweighting. Roughly speaking, the redistribution function $G_{(V,H)}$ should integrate to 0 over the support of $P_{(H|V)}^{\mathrm{MC}}$, for each value of $V$. This can be ensured with the following two conditions:

I) The essential support of $G_{(V,H)}$ is a subset of the essential support of $P_{(V,H)}^{\mathrm{PH}}$. In other words, the following is true *almost everywhere*:

$$\left( P_{(V,H)}^{\mathrm{PH}}(v,h) = 0 \quad \Longrightarrow \quad G_{(V,H)}(v,h) = 0 \right).$$
(21)

II) The redistribution does not modify quasi density of $V$, i.e.,

$$G_V(v) \equiv \int_{\mathcal{H}} \mathrm{d}h \, G_{(V,H)}(v,h) = 0, \qquad \forall v \in \mathcal{V}.$$
(22)

---

[19]This list does not include deterministic computations like performing a change of variables.

[20]Note that any quasi density, including $F_{(V,H)}^{\mathrm{ARCANE}}$, can be written down as an FCMC-able chain. However, the individual terms of this chain may not be straightforward to compute or sample from.

The rough takeaway of condition (I) is that, $G_{(V,H)}(V,H)$ should be zero for $(V,H)$ that would never be produced in the first place. Intuitively this means that $G_{(V,H)}$ should only attempt to redistribute contributions among *valid* pathways within the event generator that lead to a given event $V$.

## 2.3 Optimal Redistribution Function

For a given original event generation pipeline $P^{\mathrm{MC}}_{(W,V,H)}$, intuitively the optimal redistribution function $G^*_{(V,H)}$ is one that makes the weights of all events with the same value of $V$ be equal to each other; the different ways in which such a reweighting is indeed optimal will be discussed in Section 4.5. Concretely, $G^*_{(V,H)}$ is given by

$$G^*_{(V,H)}(v,h) \equiv \frac{F^{\mathrm{MC}}_V(v)}{P^{\mathrm{MC}}_V(v)}\, P^{\mathrm{MC}}_{(V,H)}(v,h) \,-\, F^{\mathrm{MC}}_{(V,H)}(v,h), \tag{23}$$

and the corresponding ARCANE weight $W^*_{\mathrm{ARCANE}}$ for an event with $V = v$ will exactly equal $\mathbb{E}_{\mathrm{MC}}[W \,|\, V = v]$. For completeness, the quasi density $F^{\mathrm{ARCANE\text{-}*}}_{(V,H)}$ corresponding to this choice of redistribution function is given by

$$F^{\mathrm{ARCANE\text{-}*}}_{(V,H)}(v,h) = F^{\mathrm{MC}}_V(v)\, P^{\mathrm{MC}}_{(H|V)}(h\,|\,v). \tag{24}$$

In practice, it may not be possible to construct a redistribution function $G_{(V,H)}$ that exactly matches $G^*_{(V,H)}$. However, any redistribution function $G^\dagger_{(V,H)}$ which satisfies the following condition will be as effective as $G^*_{(V,H)}$ in reducing the sign problem (proved in Section 4.3):

$$\mathrm{sgn}\!\left(F^{\mathrm{MC}}_{(V,H)}(v,h) + G^\dagger_{(V,H)}(v,h)\right) \,\in\, \left\{0\,,\, \mathrm{sgn}\!\left(F^{\mathrm{MC}}_V(v)\right)\right\}, \qquad \forall (v,h)\in \mathcal{V}\times\mathcal{H}, \tag{25}$$

where sgn is the sign function which equals $+1$, $-1$, and $0$, for positive, negative, and zero inputs, respectively. In situations where $F^{\mathrm{MC}}_V$ is non-negative, this just means that as long as the redistribution function makes $F^{\mathrm{ARCANE}}_{(V,H)}$ non-negative, it will solve the negative weights problem fully, even if it does not match $G^*_{(V,H)}$ exactly. The remaining weight-variance can be tackled using the standard unweighting technique; this is discussed further in Section 4. Note that the condition on $G^\dagger_{(V,H)}$ in (25) is significantly weaker than the condition on $G^*_{(V,H)}$ in (23)—there will typically exist a connected set of redistribution functions[21] that satisfy (25).

## 2.4 Practical Considerations

The implementation of positive resampling and other related techniques [34–38, 40, 41] discussed in Section 1.3.2 can mostly be decoupled from the implementation of the event generators, since these techniques work by postprocessing datasets produced by the event generators. In contrast, the implementation of ARCANE reweighting will involve stronger interdependence with the event generator implementations/programs, as discussed next.

### 2.4.1 Tracking Histories and Probabilities

In order to perform the reweighting as per (17), one needs the value of $(V,H)$ for each event. This involves keeping record of the outcomes of the various random decisions and samplings performed during the course of the generation of each event, even if those outcomes are not

---

[21]This set of redistribution functions will be closed under convex linear combinations (i.e., linear combinations where the coefficients are non-negative and add up to 1).

relevant beyond the event generation stage under consideration. This is similar to the record-keeping required for the techniques of Refs. [50–56], which reweight events in a given dataset to make it correspond to different values for the event-generator parameters.

Given the somewhat abstract definitions of $V$ and $H$, a natural question is, how detailed does the event record need to be? Does one need to record the outcomes of low-level random sampling operations (e.g., drawing a pseudo-random number from the uniform distribution), or can one just record the outcomes of higher-level random sampling operations (e.g., multi-channel importance sampling procedure)? There are two considerations to take into account in this context. The first is that the event record needs to be sufficiently detailed so that $P_{(V,H)}^{\text{MC}}(V,H)$ can be computed easily for each event. For example, in the veto algorithm, the high-level sampling task is to sample the next emission scale, say $T$. However, for a given value of $T$, the corresponding probability density typically cannot be computed easily. So, one will have to record the outcomes of the lower-level tasks involved in sampling $T$, namely a) sampling an emission scale *proposal* and b) randomly accepting or rejecting the proposal. Since the probabilities of the outcomes of these tasks can be computed easily, they need not be broken down further in the event record.

The second consideration is that, as previously mentioned, $(V,H)$ should cover enough of the sources of randomness in the sampling procedure so that $W$ is fully determined by $(V,H)$. As an example, in the standard or SHERPA-style multichannel importance sampling [16, 57], one need not record the channel a given event was sampled from for the sake of ARCANE reweighting, since a) the event weight is independent of the channel and b) the total probability density of an event across all channels can be computed easily. The assumption that $W$ is fully determined by $(V,H)$ is not required for the implementation of ARCANE reweighting per se; it is only required if we want to be able to *fully* eliminate the weight-variance among events with the same value of $V$ using ARCANE reweighting. This caveat is discussed further in Section 5.1.

**Subtleties in computing probabilities.**    Like event-weights in FCMC, the probability density values also evolve multiplicatively as per the chain rule $P_{(A,B)}(a,b) = P_A(a) \times P_{(B|A)}(b \,|\, a)$. However, while weights are invariant under a change of variables, probability densities acquire Jacobian factors under a change of variables. Keeping track of such Jacobian factors, making sure that the same parameterization of $V$ is used across different pathways within the event generator, etc., in order to implement ARCANE reweighting correctly can be somewhat tricky in practice.[22]   The example worked out in the companion paper, Ref. [7], illustrates such subtleties.

### 2.4.2   Constructing a Redistribution Function

The redistribution function $G_{(V,Z)}$ can be modeled in practice as a chain of functions whose integrals with respect to certain variables are known a priori. For example, one could model $G_{(V,H)}$ as

$$G_{(V,H)}^{\text{construction-eg-1}}(v,h) \equiv g^{(1)}(v,z_1) f^{(2)}(z_2 \,; v,z_1) f^{(3)}(z_3 \,; v,z_1,z_2) \dots, \tag{26}$$

where i) $(Z_1, Z_2, Z_3, \dots)$ are the hidden attributes that constitute $H$, ii) $g^{(1)}$ is a function that integrates with respect to (w.r.t.) $z_1$ to 0, and iii) $f^{(2)}$ and $f^{(3)}$ are functions that integrate to

---

[22]Typically, in the implementation of event generators, one only cares about a given Jacobian factor within a relatively localized step of the generator, in order to compute the corresponding multiplicative weight update.

1 w.r.t. $z_2$ and $z_3$, respectively. As an other example, one could model $G_{(V,H)}$ as

$$
\begin{aligned}
G_{(V,H)}^{\text{construction-eg-2}}(v,h) \equiv{}& \beta(v)\,f^{(1)}(v,z_1)\,f^{(2)}(z_2\,;\,v,z_1)\,f^{(3)}(z_3\,;\,v,z_1,z_2)\dots \\
& -\ \beta(v)\,\tilde{f}^{(1)}(v,z_1)\,\tilde{f}^{(2)}(z_2\,;\,v,z_1)\,\tilde{f}^{(3)}(z_3\,;\,v,z_1,z_2)\dots\,,
\end{aligned}
\tag{27}
$$

where i) $(Z_1, Z_2, Z_3, \dots)$ are the hidden attributes that constitute $H$, and ii) $f^{(i)}$ and $\tilde{f}^{(i)}$ are functions that integrate to 1 w.r.t. $z_i$, for $i \in \{1, 2, 3, \dots\}$. The basic idea behind these constructions is to systematically distribute an overall contribution of 0 across the various pathways, identified by $H \equiv (Z_1, Z_2, Z_3, \dots)$, that correspond to the same value of $V$. In order to complete the construction of $G_{(V,Z)}$, one still needs to specify the constituent functions, e.g., in (26) and (27). This can approached in a few different ways.

**Learning the Redistribution Function.**   One approach would be to use ML or other statistical techniques to *learn* the constituent functions of $G_{(V,H)}$ from a dataset of MC events from the original generator. Several ML architectures like normalizing flows [58] could be used to model the constituent functions in such a way that their integral is a priori known. The guiding principle behind training or adapting these ML models could be to make the resulting arcane weight $W_{\text{ARCANE}}$ in (17) approximately match the optimal value, namely $\mathbb{E}_{\text{MC}}[W \,|\, V]$. There may be several ways to approach this task. For example, the first additive term in the right-hand-side of (27) could be used to approximately subtract away the original weight of each event and the second term could be used to approximately add the optimal weight $\mathbb{E}_{\text{MC}}[W \,|\, V]$ to each event. Another approach may be to model the first term in (27) to be strictly non-negative and the second term (with the $-$ sign) to be strictly non-positive. Under this approach, one would want to train the first term to dominate the second term for events with $W < \mathbb{E}_{\text{MC}}[W \,|\, V]$ and vice versa for events with $W > \mathbb{E}_{\text{MC}}[W \,|\, V]$, so that the net effect makes $W_{\text{ARCANE}}$ roughly match the optimal value.

The technical details of this ML-based approach for constructing the redistribution function, e.g., the choice of loss function, etc. are left for future work. Note that the quality of the trained models, i.e., how close the resulting redistribution function is to $G_{(V,H)}^*$, will depend among other things on the number of datapoints used to train the models. However, the finiteness of the training dataset does not introduce a) any biases in the distribution of observable event-attributes, or b) any uncertainties that will not be accounted for by the standard analysis procedures for weighted datasets. In these ways, the limitations caused by finiteness of the training dataset are similar in both ARCANE reweighting and importance sampling (implemented, say, via a VEGAS grid adapted or trained with a finite dataset).

The dataset used to train the redistribution function will contain the values of $V$, $H$, $W$, and $P_{(V,H)}^{\text{MC}}(V,H)$ for each event. The dataset may also contain the values of $P_X^{\text{MC}}(X)$, where $X$ represents a subset of the information in $(V,H)$. Note that not all of this information will be available to the subsequent simulation and analysis stages in a typical HEP-workflow. Consequently, the data processing inequality cannot be used to argue against the usefulness of ARCANE reweighting trained using a finite dataset.

**Engineering the Redistribution Function.**   Instead of learning $G_{(V,H)}$ from data, one may also be able to engineer a good redistribution function, by leveraging one's knowledge about the original event generation pipeline. This is the approach taken in the companion paper Ref. [7]. A potential advantage of engineering redistribution functions over learning them from data is the following: Under the learning approach, one needs to train $G_{(V,H)}$ using data separately for every choice of values for the event-generator-parameters (or train an appropriately parameterized redistribution function). On the other hand, the engineering approach

may lead to an easier way to construct $G_{(V,H)}$ for different choices of generator-parameter-values.

As a simple illustration of engineering $G_{(V,H)}$, consider the example $F_V^{\text{PH-eg-1}}$ in (4), where each $V$ can be produced by two possible pathways indicated by $Z \in \{1, 2\}$. After producing an event with $V = \nu$ from either pathway, let us say one has the ability to enumerate and "walk through" both generator pathways that lead to $V = \nu$. Now, $G_{(V,Z)}^*$ from (23) can be constructed easily for this example as follows:

$$G_{(V,Z)}^{\text{eg-1-}*}(\nu, z) \; \equiv \; P_{(V,Z)}^{\text{MC}}(\nu, z) \left[ \frac{\displaystyle\sum_{z'=1}^{2} P_{(V,Z)}^{\text{MC}}(\nu, z') \, W_{\text{MC-func}}(\nu, z')}{\displaystyle\sum_{z'=1}^{2} P_{(V,Z)}^{\text{MC}}(\nu, z')} \; - \; W_{\text{MC-func}}(\nu, z) \right], \quad (28)$$

where $W_{\text{MC-func}}$ is a function that returns the original weight of an event with a given value of $(V, Z)$. Every term in this expression can be computed if one has the ability to loop through the alternative MC histories that lead to the same $V$. The ARCANE weight resulting from this choice of redistribution function is given by

$$W_{\text{ARCANE-func}}^{\text{eg-1-}*}(\nu, z) \; \equiv \; \frac{\displaystyle\sum_{z'=1}^{2} P_{(V,Z)}^{\text{MC}}(\nu, z') \, W_{\text{MC-func}}(\nu, z')}{\displaystyle\sum_{z'=1}^{2} P_{(V,Z)}^{\text{MC}}(\nu, z')} \qquad \text{(independent of } z\text{)}. \quad (29)$$

This is similar to how events are weighted under the standard or SHERPA-style multichannel importance sampling—$Z$ represents the different channels through which one can produce $V$, and one iterates through all the channels to weight each event (even though each event is produced only from one of the channels).

One can begin implementing ARCANE reweighting for MC@NLO event generation along similar lines, treating the different $\mathbb{H}$- and $\mathbb{S}$-type pathways (with different choices for emitter–spectator pairings) as being different channels for sampling resolved events. There are other, possibly continuous, latent attributes (e.g., from the veto algorithm) that make the MC@NLO situation more complicated than the example described here; these are discussed and handled in the companion paper, Ref. [7]. While in the simple example considered here, it was possible to construct the optimal redistribution function, in more complicated situations one may have to settle for constructing a good, but not optimal, redistribution function.

## 3 Interplay Between ARCANE and Rejection Reweightings

Rejection reweighting is a variance reduction technique commonly used in HEP. This section briefly reviews this technique and describes how ARCANE and rejection reweighting techniques can be used in tandem with each other.

### 3.1 Rejection Reweighting and Unweighting Techniques

Rejection reweighting operates by probabilistically accepting or rejecting each event produced by an event generation pipeline, with some event-specific acceptance probability $\alpha$; the rejected events are not processed further. The weights of the accepted events are appropriately scaled to compensate for this downsampling and keep the quasi density of the event-attributes

unchanged. There are a few different ways to formalize this technique;[23] for simplicity, let the production of a single event, including the rejection reweighting step, proceed as follows:

1. A candidate event is produced using a given MC pipeline with weight $W$.

2. The candidate event is passed through the accept/reject filter with an event-specific acceptance probability $\alpha \in [0, 1]$, which can depend on $W$ as well as other event-attributes. All events with $W > 0$ should satisfy $\alpha > 0$. It will be assumed in this paper that $\alpha$ is uniquely determined by Monte Carlo history of the event leading up to the rejection reweighting step.[24]

   (a) If the candidate event is rejected, one discards it and restarts the sampling procedure from step 1.

   (b) If the candidate event is accepted, it is returned as the sampled event, with its weight $W$ rescaled as follows:

$$W \longmapsto \frac{\epsilon\, W}{\alpha}, \qquad \text{or, more explicitly,} \qquad W_{\text{post-rej-rwt}} \equiv \frac{\epsilon\, W_{\text{pre-rej-rwt}}}{\alpha}, \qquad (30)$$

   where $\epsilon$ is the average acceptance probability, which is given by the expected value of the random variable $\alpha$. $\epsilon$ is referred to as the efficiency of the rejection reweighting step.

It is easy to verify that this procedure preserves the quasi density of any event-attribute $X$ that is not changed by the rejection reweighting step (this includes all visible and hidden attributes of the event, but not $W$). In practice, $\epsilon$ is typically not known exactly, so an MC estimate $\hat{\epsilon}$ is used instead of $\epsilon$ in (30). Also, for operational convenience, constant terms like $\hat{\epsilon}$, $\widehat{\mathbb{E}}_{\text{MC}}\big[|W|\big]$, etc. are often factored out of the event-weight, and directly incorporated in the subsequent analyses as needed.[25] For pedagogical simplicity, such complications will be ignored here.

The rejection reweighting technique can be used to alter the weight-distribution of the accepted MC events, which undergo the subsequent stages of the event production pipeline. However, this comes with the additional computational cost associated with the generation of the eventually rejected candidate events, which increases the amortized time and resource requirements per accepted event. Typically, the rejection reweighting technique is performed before the detector simulation stage, so the increased costs are only in the particle-level event generation.

Let us say one with working with a constraint of the form $\epsilon \geq \epsilon_{\text{lb}}$ on the average acceptance probability $\epsilon$. It can be shown that for (i) any $\epsilon_{\text{lb}} \in (0, 1)$ and (ii) any convex function $f_{\text{convex}}$ defined on non-negative reals satisfying $f_{\text{convex}}(0) < \infty$, $\mathbb{E}_{\text{post-rej-rwt}}\big[f_{\text{convex}}(|W_{\text{post-rej-rwt}}|)\big]$ is minimized by choosing $\alpha$ as follows:

$$\alpha_{\text{opt}} \equiv \min\left(1, \left|\frac{W_{\text{pre-rej-rwt}}}{W_{\text{ref}}}\right|\right), \qquad (31)$$

---

[23]For example, one can consider the rejected events as being zero-weight events as opposed to discarded events. As another example, one can treat the number of events either before or after the downsampling as being predetermined. The statistical analysis for these different cases will be subtly, and typically inconsequentially, different.

[24]This assumption is not necessary for the validity of the rejection reweighting technique—in principle, the value of $\alpha$ could itself have an additional source of randomness.

[25]Since the same value of $\hat{\epsilon}$ is used in all event-weights, the error in $\hat{\epsilon}$ induces a small correlation between the events; this effect will be negligible in the large number of samples limit. Factoring out $\hat{\epsilon}$ and incorporating it directly in the subsequent analyses (along with statistical uncertainty estimates) is one way to handle the aforementioned correlation. Additional subtle correlations and/or biases can also be introduced if $\hat{\epsilon}$ is not estimated using a separate independent dataset; this effect will also become negligible in the large number of samples limit.

where $|W_{\text{ref}}|$ is a positive constant that determines $\epsilon$ (or is determined by $\epsilon_{\text{lb}}$); a higher value of $|W_{\text{ref}}|$ corresponds to lower value of $\epsilon$. As a special case, setting $f_{\text{convex}}(|W|) = W^2$, it can be seen that this choice of acceptance function minimizes the weight-variance among the selected events, for a given $\epsilon_{\text{lb}}$.

**Unweighting** is a special case of rejection reweighting. It involves choosing $\alpha$ as per (31), with $|W_{\text{ref}}|$ set to an essential upper bound[26] of the $|W_{\text{pre-rej-rwt}}|$. In this case, a) $\alpha$ will be directly proportional to $|W_{\text{pre-rej-rwt}}|$, b) $|W_{\text{post-rej-rwt}}|$ will be a constant given by $\mathbb{E}_{\text{MC}}\big[|W_{\text{pre-rej-rwt}}|\big]$. c) $\epsilon$ will be given by $\mathbb{E}_{\text{MC}}\big[|W_{\text{pre-rej-rwt}}|\big] / |W_{\text{ref}}|$. In this case, $\epsilon$ is referred to as the **unweighting efficiency**.

## 3.2 Limitations and Constraints of Rejection Reweighting

Rejection reweighting cannot change the sign of the weights of accepted events;[27] this sets a floor to the weight-variance achievable by the technique, as discussed in Section 4.2. Furthermore, for **any** event attribute $X$ (physically motivated or otherwise) that is unaffected by the rejection reweighting procedure, the functions

$$F_X^{\text{MC}}(x) \equiv P_X^{\text{MC}}(x)\, \mathbb{E}_{\text{MC}}\big[W \,\big|\, X = x\big] \qquad \text{and} \qquad P_X^{\text{MC}}(x)\, \mathbb{E}_{\text{MC}}\big[|W| \,\big|\, X = x\big] \qquad (32)$$

will be unaffected by the rejection reweighting procedure.[28] From these, it can be shown that the following quantities are also unaffected by rejection reweighting:

$$P_X^{\text{MC}}(x)\, \mathbb{E}_{\text{MC}}\big[\max(0, W) \,\big|\, X = x\big] \;\equiv\; \frac{P_X^{\text{MC}}(x)}{2}\, \mathbb{E}_{\text{MC}}\Big[|W| + W \,\Big|\, X = x\Big], \qquad (33)$$

$$P_X^{\text{MC}}(x)\, \mathbb{E}_{\text{MC}}\big[\max(0, -W) \,\big|\, X = x\big] \;\equiv\; \frac{P_X^{\text{MC}}(x)}{2}\, \mathbb{E}_{\text{MC}}\Big[|W| - W \,\Big|\, X = x\Big]. \qquad (34)$$

A special case of these results, with $X$ chosen to be a constant event-attribute, is that the quantities $\mathbb{E}_{\text{MC}}[W]$, $\mathbb{E}_{\text{MC}}\big[|W|\big]$, $\mathbb{E}_{\text{MC}}\big[\max(0, W)\big]$, and $\mathbb{E}_{\text{MC}}\big[\max(0, -W)\big]$ are invariant under rejection reweighting.

## 3.3 Complementarity of ARCANE and Rejection Reweighting Techniques

ARCANE reweighting and rejection reweighting both modify the distribution of event-weights without changing the quasi density of the visible attributes. However, they accomplish this task using very different mechanisms. ARCANE reweighting modifies the event-weight

$$W \equiv \frac{F_{(V,H)}(V, H)}{P_{(V,H)}(V, H)}$$

by modifying the numerator $F_{(V,H)}(V, H)$, which captures the "contribution" of the pathway represented by $(V, H)$ to the visible event $V$. On the other hand, rejection reweighting affects the event-weights by modifying the denominator, namely the sampling probability density $P_{(V,H)}(V, H)$.

Due to the differences in the reweighting mechanism, the two techniques also have different limitations. As discussed before, rejection reweighting cannot modify the quasi density

---

[26]In practice, one might set $|W_{\text{ref}}|$ to the sample maximum of $|W_{\text{pre-rej-rwt}}|$, possibly multiplied by some factor greater than 1. If this sample maximum is not estimated using a separate independent dataset, a small correlation would be induced between the events; this effect is negligible in the large-number-of-samples limit.

[27]This means that the sign of $W$ can be considered as a latent event attribute unaffected by rejection reweighting. This is relevant for Footnote 28.

[28]The invariance of the latter can be seen directly or be derived from the invariance of the former as follows, by treating $S_W \equiv \text{sgn}(W)$ as an event attribute: $P_X^{\text{MC}}(x)\, \mathbb{E}_{\text{MC}}\big[|W| \,\big|\, X = x\big] \;=\; F_{(X,S_W)}^{\text{MC}}(x, +1) - F_{(X,S_W)}^{\text{MC}}(x, -1)$.

$F_{(V,H)}^{\mathrm{MC}}$ (in particular its sign). So, it cannot reduce the weight-variance stemming from the negative weights problem. On the other hand, ARCANE reweighting cannot change $P_V^{\mathrm{MC}}$ and $\mathbb{E}_{\mathrm{MC}}[W \mid V]$. In other words, ARCANE reweighting cannot eliminate the weight-variance arising from $V$ being under- or over-sampled in certain regions of the phase space. ARCANE reweighting is not intended to be an alternative to rejection reweighting. Rather, the proposal is to employ both techniques to eliminate different components of the weight-variance and improve sample efficiency; this is discussed further in Section 4. The mechanics of incorporating both techniques into an event generation pipeline are briefly described next.

### 3.4 Incorporating Both ARCANE and Rejection Reweighting Techniques

As a simple case, the ARCANE reweighting and rejection reweighting techniques can be incorporated into a generation pipeline successively. Alternatively, after performing one of the reweighting techniques, one could perform several other event generation steps before performing the other reweighting technique. In general, the techniques can be employed multiple times, in any order, in different parts of the generation pipeline, as needed.

Recall that in order to perform ARCANE reweighting, one needs to track the probability density of $(V,H)$. Tracking this probability density in the presence of rejection reweighting steps can be done as follows. Let $T$ represent all the attributes of an event generated before a rejection reweighting step is performed. After performing the rejection reweighting, the probability density of an accepted event $T$ is given by:

$$P_T^{\text{post-rej-rwt}}(T) \equiv \frac{\alpha \, P_T^{\text{pre-rej-rwt}}(T)}{\epsilon} \,, \tag{35}$$

where $\alpha$ is the acceptance probability, which is fully determined by the event "tape" $T$, and $\epsilon$ has the same definition as before. The probability density $P_T^{\text{post-rej-rwt}}$ should be used for the subsequent tracking of probability densities, which will eventually be used in the ARCANE reweighting formulas in Section 2.

In the simple case, when the two reweighting techniques are incorporated successively, regardless of the order of the techniques, the final weight of an event is given by

$$W_{\mathrm{final}} \equiv \frac{\epsilon}{\alpha} \left[ W_{\mathrm{orig}} + \frac{G_{(V,H)}(V,H)}{P_{(V,H)}^{\mathrm{MC}}(V,H)} \right], \tag{36}$$

where $W_{\mathrm{orig}}$ is the original weight of the event before any reweighting and $P_{(V,H)}^{\mathrm{MC}}$ is the sampling probability density of $(V,H)$ under the original event generation procedure (prior to applying the accept-reject filter). Here $\epsilon$, $\alpha$, and $G_{(V,H)}$ have the same definitions as before. Note that performing ARCANE reweighting after unweighting will result in event-weights not having the same magnitudes (barring some trivial cases). For this reason, it may be preferable to apply the unweighting technique after performing ARCANE reweighting, when the two techniques are intended to be applied successively.

## 4 Quantitative Analysis and Discussions

An appropriate performance metric (albeit difficult to quantify and estimate) for evaluating MC event production pipelines is the "overall time and resource requirements[29] of an event

---

[29]Overall time and resource requirement (a) is not a single number, and can be quantified in different ways, and (b) depends on various factors like computing hardware architecture, parallelizability of the event production pipeline, etc. Such complications are ignored in the discussions in this paper.

production pipeline for reaching a certain target precision or sensitivity[30] in an experimental analysis." Metrics like unweighting efficiency, fraction of negative events (post unweighting), etc. capture certain aspects of this overall performance metric. Incorporating ARCANE reweighting into an existing event generation pipeline brings additional costs associated with a) constructing the redistribution function $G_{(V,H)}$ as a preliminary step, b) tracking and recording hidden event attributes, c) computing the additive reweighting term, etc., in addition to the time and effort involved in implementing the technique in software. These costs will depend on the specific use-case and implementation details, so a complete cost–benefit analysis cannot be performed here. However, this section provides a few general arguments regarding the potential computational benefits of ARCANE reweighting to the HEP event production pipeline.

**Notation.** In Section 2 and Section 3, subscripts were added to event weights to simplify the presentation of the reweighting formulas. In the rest of this paper, descriptive subscripts (e.g. "MC-plus-rej-rewgting") will be added to $\mathbb{E}$, Var, etc. to denote the specifics of the event generation and weighting procedure and no subscripts will be used with $W$ (unless necessary), like for instance $\mathbb{E}_{\text{MC-plus-rej-rewgting}}[W]$. This is the original convention used in Section 1 and is better suited for the quantitative discussions in this section.

## 4.1 Preliminaries

The time and resource requirements to reach a given sensitivity can be analyzed in two parts— a) the amortized time and resource requirements per sampled event, and b) the total number of simulated events needed to reach the desired sensitivity, under a given event generation and analysis pipeline.

**Amortized resource requirements per sampled event**

*Simple treatment:* In many situations of interest, detector simulation (along with the subsequent stages of the simulation pipeline) is much more computationally expensive than the particle-level event generation, even after accounting for the downsampling involved in rejection reweighting (for typical values of acceptance rates). If one assumes that this imbalance will continue to hold even after adding ARCANE reweighting to the particle-level event generation, then the amortized resource requirements per detector-level event will be approximately independent of the several details of the event generation pipeline, including (i) whether ARCANE reweighting is performed, (ii) value of the unweighting efficiency (if performed), etc.[31]

*More complex treatment:* In some situations of interest, like the simulation of events with a high-multiplicity of final state particles from the hard-scattering, the calculation of matrix elements can be sufficiently expensive and the unweighting efficiency can be sufficiently low that the resource requirements for particle-level generation can be comparable to those of the detector simulation stage. In this case, the amortized resource requirements will depend significantly on a number of aspects of the generation pipeline. Two factors relevant in the context of this paper are (a) the efficiency of rejection reweighting (if used), and (b) the per-event resource requirements associated with ARCANE reweighting. As mentioned earlier, the latter will not be quantified or estimated in this paper.

---

[30]The target sensitivity is, of course, not a fixed quantity in practice, and depends on the cost–benefit trade-off of various factors.

[31]In principle, detector simulation will require different amount of resources for different events within a given dataset. An assumption here is that these generator-details will not alter the sampling distribution of events in such a way that the average detector-simulation-resource-requirements per event is affected appreciably.

**Total number of simulated events needed**

*Simple treatment:* For simplicity, the total number of events required, is often taken to be monotonically related to the variance of event-weights, or $\mathbb{CV}^2_{\text{MC}}[W]$ (square of the coefficient of variation of $W$) defined as[32]

$$\mathbb{CV}^2_{\text{MC}}[W] \equiv \frac{\text{Var}_{\text{MC}}[W]}{\mathbb{E}^2_{\text{MC}}[W]}. \tag{37}$$

Lower values of $\mathbb{CV}^2_{\text{MC}}[W]$ roughly corresponds to fewer simulated events needed.

*More complex treatment:* Various factors influence the sensitivity of an experimental analysis. The precisions of various Monte Carlo predictions (e.g., for the expected counts in different histogram bins) only constitute one such factor. Different regions in the phase space (of the reconstructed event attributes) (i) may be important to a particular analysis to different degrees, (ii) may have different degrees of systematic uncertainties, (iii) may be intentionally over- or under-sampled [2,59], taking into account (i) and (ii), etc. So, $\mathbb{CV}^2_{\text{MC}}[W]$ may not necessarily be monotonically related to the number of events needed by an analysis. To facilitate a more nuanced discussion, let a local or conditional $\mathbb{CV}^2_{\text{MC}}$ be defined as follows:

$$\mathbb{CV}^2_{\text{MC}}\big[W \,\big|\, X = x\big] \equiv \frac{\text{Var}_{\text{MC}}[W \,|\, X = x]}{\mathbb{E}^2_{\text{MC}}[W \,|\, X = x]}, \tag{38}$$

where $X$ is an arbitrary event-attribute.

Let MC and MC$'$ be two Monte Carlo procedures for sampling events from the same physics model, i.e., $F_V^{\text{MC}} \equiv F_V^{\text{MC}'}$. The following statement holds any classic fully-histogram-based[33] experimental analysis, under the subsequent list of assumptions:

**Statement 1.** *If $P_V^{\text{MC}} \equiv P_V^{\text{MC}'}$, $F_V^{\text{MC}} \equiv F_V^{\text{MC}'}$, and $\mathbb{CV}^2_{\text{MC}}\big[W \,\big|\, V = v\big] \leq \mathbb{CV}^2_{\text{MC}'}\big[W \,\big|\, V = v\big]$ for all $v \in \mathcal{V}$, then the number of events needed to reach a target sensitivity under MC will be no greater than the number of events needed under MC$'$.*

*Assumptions:*

- *The subsequent processing of each event (including remaining stages of the event production pipeline) depends only on $V$, with weights evolving multiplicatively.*

- *No event reweighting or downsampling techniques are involved in the subsequent processing of the event (i.e., any reweighting/downsampling techniques are included in MC and MC$'$ themselves).*

- *All aspects of the analysis, including, e.g., devising the event-selection criteria, is performed using histograms.*

- *The non-normality of the errors in Monte Carlo predictions for expected histogram-counts do not play a significant role.*

□

---

[32]Often event-weights are rescaled by overall factors, which are accounted for elsewhere. $\mathbb{CV}^2_{\text{MC}}[W]$ is invariant under such a rescaling.

[33]This statement could roughly be expected to hold even for non-classic analyses that use, say, neural networks. However, the behavior of neural-network-training (using complicated meta-heuristic optimization algorithms) under different local weight distributions cannot be guaranteed.

The rationale behind the statement is that, for the same total number of events, the variances of the Monte Carlo predictions (for different histogram bins) under MC would be no greater than those under MC′. This rationale can be quantified using the notion of "effective number density". Let $N_{\text{tot}}$ be the total number of events in a dataset, generated using an even generation pipeline "MC". One can define the actual and effective number densities for an event-attribute $X$ as follows:

$$n_{\text{MC}}(X = x) \quad \equiv \quad N_{\text{tot}} \, P_X^{\text{MC}}(x), \tag{39}$$

$$n_{\text{MC}}^{(\text{eff})}(X = x) \quad \equiv \quad \frac{N_{\text{tot}} \, P_X^{\text{MC}}(x)}{\mathbb{E}_{\text{MC}}\big[W^2 \mid X = x\big] / \mathbb{E}_{\text{MC}}^2\big[W \mid X = x\big]} \tag{40}$$

$$= \quad \frac{N_{\text{tot}} \, P_X^{\text{MC}}(x)}{1 + \mathbb{CV}_{\text{MC}}^2\big[W \mid X = x\big]} . \tag{41}$$

If $X$ is a discrete event attribute (like a histogram bin-index), then $n_{\text{MC}}^{(\text{eff})}(X = x)$ roughly captures the number density $n_{\text{MC}'}(X = x)$ one would need under an alternative sampling procedure MC′, satisfying a) $F_X^{\text{MC}'} \equiv F_X^{\text{MC}}$ and b) $\mathbb{CV}_{\text{MC}'}^2[W \mid X = x] = 0$, in order to reach the same precision in the Monte Carlo estimate for $F_X^{\text{MC}}(x)$. By choosing, $X$ to be a constant-event attribute here, one can define an overall effective number of events as follows:

$$N_{\text{MC}}^{(\text{eff})} \quad \equiv \quad \frac{N_{\text{tot}}}{1 + \mathbb{CV}_{\text{MC}}^2[W]} . \tag{42}$$

## 4.2 Quantifying the Negative Weights Problem

There are two related ways to quantify the negative weights problem. The first is as a fundamental limitation in modeling a target quasi density of indeterminate sign. The second is as a lower-bound on the weight-variance achievable using rejection reweighting.

**Lowest variance achievable for a given target quasi density.** Using the law of total variance [60] and the fact that $\text{Var}[X] = \text{Var}\big[|X|\big] + \mathbb{E}^2\big[|X|\big] - \mathbb{E}^2[X]$ for any random variable $X$, one can write

$$\text{Var}_{\text{MC}}[W] = \mathbb{E}_{\text{MC}}\Big[\text{Var}_{\text{MC}}\big[W \mid X\big]\Big] + \text{Var}_{\text{MC}}\Big[\mathbb{E}_{\text{MC}}\big[W \mid X\big]\Big] \tag{43}$$

$$= \underbrace{\mathbb{E}_{\text{MC}}\Big[\text{Var}_{\text{MC}}\big[W \mid X\big]\Big] + \text{Var}_{\text{MC}}\Big[\big|\mathbb{E}_{\text{MC}}\big[W \mid X\big]\big|\Big]}_{\geq \, 0} + \mathbb{E}_{\text{MC}}^2\Big[\big|\mathbb{E}_{\text{MC}}\big[W \mid X\big]\big|\Big] - \mathbb{E}_{\text{MC}}^2[W], \tag{44}$$

where $X$ is an arbitrary event-attribute. This leads to the following inequality:

$$\mathbb{CV}_{\text{MC}}^2[W] \equiv \frac{\text{Var}_{\text{MC}}[W]}{\mathbb{E}_{\text{MC}}^2[W]} \geq \mathbb{SP}_{\text{MC}}[X] \equiv \frac{\mathbb{E}_{\text{MC}}^2\Big[\big|\mathbb{E}_{\text{MC}}\big[W \mid X\big]\big|\Big] - \mathbb{E}_{\text{MC}}^2[W]}{\mathbb{E}_{\text{MC}}^2[W]} \tag{45}$$

$$= \frac{\left(\int_{\mathcal{X}} dx \, \big|F_X^{\text{MC}}(x)\big|\right)^2 - \left(\int_{\mathcal{X}} dx \, F_X^{\text{MC}}(x)\right)^2}{\left(\int_{\mathcal{X}} dx \, F_X^{\text{MC}}(x)\right)^2}, \tag{46}$$

with equality if and only if $P_X^{\text{MC}}$ is proportional to $|F_X^{\text{MC}}|$ and $W$ is fully determined by $X$. Note that this lower-bound on $\mathbb{CV}_{\text{MC}}^2[W]$, namely $\mathbb{SP}_{\text{MC}}[X]$, depends only on the quasi density $F_X^{\text{MC}}$. In other words, $\mathbb{SP}_{\text{MC}}[X]$ is a lower-bound on $\mathbb{CV}_{\text{MC}'}^2[W]$ for any Monte Carlo procedure MC′

such that $F_X^{\text{MC}'} \equiv F_X^{\text{MC}}$; it is a fundamental limitation of modeling a given target quasi density. Furthermore, $\mathbb{SP}_{\text{MC}}[X]$ is 0 if $F_X^{\text{MC}}$ is non-negative (or non-positive) everywhere. So, $\mathbb{SP}_{\text{MC}}[X]$ can be thought of as a measure of the Sign Problem inherent in the quasi density or distribution $F_X^{\text{MC}}$.

Different choices of event attributes lead to different $\mathbb{SP}_{\text{MC}}$-s. It can be shown that for two event-attributes $X$ and $Y$ such that $X$ is fully determined by $Y$, $\mathbb{SP}_{\text{MC}}[X] \leq \mathbb{SP}_{\text{MC}}[Y]$. This implies that

$$\mathbb{SP}_{\text{MC}}[V,H] \equiv \mathbb{SP}_{\text{MC}}\big[V, H_{\text{thr}}, H_{\text{imp}}\big] \geq \mathbb{SP}_{\text{MC}}[V, H_{\text{thr}}] \geq \mathbb{SP}_{\text{MC}}[V], \tag{47}$$

where, as before, $H_{\text{thr}}$ and $H_{\text{imp}}$ are the hidden variables originating from the theory description of $F_V^{\text{MC}}$ and the Monte Carlo implementation, respectively.

In principle, $\mathbb{SP}_{\text{MC}}[V,H]$ could be strictly greater than $\mathbb{SP}_{\text{MC}}[V,H_{\text{thr}}]$, if the Monte Carlo implementation worsens the sign problem present in the theory description/formalism. For example, in the veto algorithm for sampling parton shower emissions, if the auxiliary kernel functions (used to sample emission proposals) are not guaranteed to be greater than or equal to the corresponding splitting kernel functions, then one would be introducing an additional sign problem into the generation pipeline. However, such inefficiencies are usually avoided in practice, so in typical situations, $\mathbb{SP}_{\text{MC}}[V,H]$ equals $\mathbb{SP}_{\text{MC}}[V,H_{\text{thr}}]$.

**Lowest variance achievable by rejection reweighting.**   Since the rejection reweighting technique preserves the quasi density of $(V,H)$, it is subject to the following constraint:

$$\mathbb{CV}^2_{\text{MC-plus-rej-rewgtng}}[W] \geq \mathbb{SP}_{\text{MC}}[V,H] \equiv \frac{\mathbb{E}^2_{\text{MC}}[|W|] - \mathbb{E}^2_{\text{MC}}[W]}{\mathbb{E}^2_{\text{MC}}[W]}. \tag{48}$$

The lower-bound in (48) will be achieved by the special case of rejection reweighting, namely the unweighting technique, i.e.,

$$\mathbb{CV}^2_{\text{MC-plus-unweighting}}[W] = \mathbb{SP}_{\text{MC}}[V,H]. \tag{49}$$

So, $\mathbb{SP}_{\text{MC}}[V,H]$ can be thought of as a measure of the sign problem in the event generation pipeline; it captures the component of the weight-variance that cannot be eliminated via unweighting. Let $f_{\text{MC}}^{(\text{neg})}$ represent the fraction of events with negative weights, if one performs an unweighting procedure after sampling events using MC:

$$f_{\text{MC}}^{(\text{neg})} \equiv \frac{1}{2} \left( \frac{\mathbb{E}_{\text{MC}}\big[|W| - W\big]}{\mathbb{E}_{\text{MC}}\big[|W|\big]} \right) = \frac{\mathbb{E}_{\text{MC}}\big[\max(0, -W)\big]}{\mathbb{E}_{\text{MC}}\big[|W|\big]}. \tag{50}$$

$\mathbb{SP}_{\text{MC}}[V,H]$ is related to $f_{\text{MC}}^{(\text{neg})}$ as follows:

$$1 + \mathbb{SP}_{\text{MC}}[V,H] = \left(1 - 2 f_{\text{MC}}^{(\text{neg})}\right)^{-2}. \tag{51}$$

The closer $f_{\text{MC}}^{(\text{neg})}$ is to 1/2, the worse the sign problem in the Monte Carlo pipeline "MC".

**Local measures of the sign problem.**   One can define local variants of $\mathbb{SP}$ and $f^{(\text{neg})}$ as follows:

$$\mathbb{SP}_{\text{MC}}\big[X \mid Y = y\big] \equiv \frac{\mathbb{E}^2_{\text{MC}}\Big[\big|\mathbb{E}_{\text{MC}}\big[W \mid X, Y = y\big]\big| \ \Big| \ Y = y\Big] - \mathbb{E}^2_{\text{MC}}\big[W \mid Y = y\big]}{\mathbb{E}^2_{\text{MC}}\big[W \mid Y = y\big]} \tag{52}$$

$$= \frac{\left(\int_{\mathcal{X}} dx \left| F_{(X,Y)}^{\mathrm{MC}}(x,y) \right| \right)^2 - \left(\int_{\mathcal{X}} dx \, F_{(X,Y)}^{\mathrm{MC}}(x,y) \right)^2}{\left(\int_{\mathcal{X}} dx \, F_{(X,Y)}^{\mathrm{MC}}(x,y) \right)^2}, \tag{53}$$

$$f_{\mathrm{MC}}^{(\mathrm{neg,local})}(Y = y) \equiv \frac{1}{2} \left( \frac{\mathbb{E}_{\mathrm{MC}}\left[ |W| - W \mid Y = y \right]}{\mathbb{E}_{\mathrm{MC}}\left[ |W| \mid Y = y \right]} \right) = \frac{\mathbb{E}_{\mathrm{MC}}\left[ \max(0, -W) \mid Y = y \right]}{\mathbb{E}_{\mathrm{MC}}\left[ |W| \mid Y = y \right]} \tag{54}$$

for arbitrary event attributes $X$ and $Y$. Analogous to the earlier equations, the following properties hold for local variants of $\mathbb{CV}$, $\mathbb{SP}$, and $f^{(\mathrm{neg})}$:

$$\mathbb{SP}_{\mathrm{MC}}\left[V, H \mid X = x\right] \geq \mathbb{SP}_{\mathrm{MC}}\left[V, H_{\mathrm{thr}} \mid X = x\right] \geq \mathbb{SP}_{\mathrm{MC}}\left[V \mid X = x\right], \tag{55}$$

$$\mathbb{CV}_{\mathrm{MC}}^2\left[W \mid X = x\right] \geq \mathbb{SP}_{\mathrm{MC}}\left[V, H \mid X = x\right], \tag{56}$$

$$\mathbb{CV}_{\mathrm{MC\text{-}plus\text{-}rej\text{-}rewgtng}}^2\left[W \mid X = x\right] \geq \mathbb{SP}_{\mathrm{MC}}\left[V, H \mid X = x\right], \tag{57}$$

$$\mathbb{CV}_{\mathrm{MC\text{-}plus\text{-}unweighting}}^2\left[W \mid X = x\right] = \mathbb{SP}_{\mathrm{MC}}\left[V, H \mid X = x\right], \tag{58}$$

$$1 + \mathbb{SP}_{\mathrm{MC}}\left[V, H \mid X = x\right] = \left(1 - 2 f_{\mathrm{MC}}^{(\mathrm{neg,local})}(X = x)\right)^{-2}. \tag{59}$$

**Scaling of effective number density for unweighted samples.** From (39), (41), (56), (58), and (59), one can see that

$$\frac{n_{\mathrm{MC}}^{(\mathrm{eff})}(X = x)}{n_{\mathrm{MC}}(X = x)} \leq \frac{n_{\mathrm{MC\text{-}plus\text{-}unweighting}}^{(\mathrm{eff})}(X = x)}{n_{\mathrm{MC\text{-}plus\text{-}unweighting}}(X = x)} \tag{60}$$

$$= \frac{1}{1 + \mathbb{SP}_{\mathrm{MC}}\left[V, H \mid X = x\right]} = \left(1 - 2 f_{\mathrm{MC}}^{(\mathrm{neg,local})}(X = x)\right)^2. \tag{61}$$

This points to the fact that the effective number density increases with the actual number density at a slower rate for regions of the phase space with worse sign problem. This means that errors in different regions or histogram bins will reduce (with number density) at different rates for simulated and experimental datasets.[34] Such a discrepancy is often not desirable in practice. Fortunately, some of this discrepancy is offset by the fact that when one performs an unweighting procedure, regions with worse negative weights problem naturally get oversampled. Concretely, using the results in Section 3.2, the probability density of an event-attribute $X$ after unweighting is given by

$$P_X^{\mathrm{MC\text{-}plus\text{-}unweighting}}(x) = \frac{P_X^{\mathrm{MC}}(x) \, \mathbb{E}_{\mathrm{MC}}\left[ |W| \mid X = x \right]}{\mathbb{E}_{\mathrm{MC}}\left[ |W| \right]} \tag{62}$$

$$= \underbrace{\left[ \frac{F_X^{\mathrm{MC}}(x)}{\int_{\mathcal{X}} dx \, F_X^{\mathrm{MC}}(x)} \right]}_{\substack{\text{distribution under completely} \\ \text{unsigned and unweighted sampling,} \\ \text{assuming } F_X^{\mathrm{MC}} \text{ is non-negative}}} \left[ \frac{1 - 2 f_{\mathrm{MC}}^{(\mathrm{neg})}}{1 - 2 f_{\mathrm{MC}}^{(\mathrm{neg,local})}(X = x)} \right]. \tag{63}$$

This shows that regions with a worse local sign problem (when compared to the overall sign problem) are oversampled post-unweighting. Plugging this into (41) leads to

$$n_{\mathrm{MC\text{-}plus\text{-}unweighting}}^{(\mathrm{eff})}(X = x) = \left[ \frac{N_{\mathrm{tot}} \, F_X^{\mathrm{MC}}(x)}{\int_{\mathcal{X}} dx \, F_X^{\mathrm{MC}}(x)} \right] \left(1 - 2 f_{\mathrm{MC}}^{(\mathrm{neg})}\right) \left(1 - 2 f_{\mathrm{MC}}^{(\mathrm{neg,local})}(X = x)\right). \tag{64}$$

---

[34]More precisely, hypothetical experimental datasets corresponding to the physics model being simulated.

This shows that even after unweighting, some discrepancy still persists in the rates at which uncertainties reduce in different regions or histogram bins. This discrepancy can be corrected with a more nuanced biased sampling, but such a biasing will be dependent on the analysis-level variable being analyzed, say, using histograms. An analysis-agnostic correction of the discrepancy being discussed here is not possible, except in the trivial case where $f_{\mathrm{MC}}^{(\mathrm{neg,local})}(V = v)$ is a constant. So, in practice, the impact of negative weights on the simulation requirements could be worse than one might expect from $N_{\mathrm{MC}}^{(\mathrm{eff})}$ in (42), depending on the value of $f_{\mathrm{MC}}^{(\mathrm{neg,local})}$ in different regions in the phase space of the final reconstruction-level event.

### 4.3 Effect of ARCANE Reweighting on the Sign Problem Metric $\mathbb{SP}$

Since ARCANE reweighting does not preserve the quasi density of $(V, H)$, it can be used to reduce the sign problem below $\mathbb{SP}_{\mathrm{MC}}[V, H]$. Let "MC-plus-ARCANE-†" represent a Monte Carlo procedure which involves adding ARCANE reweighting, with a redistribution function $G_{(V,H)}^{\dagger}$ satisfying the condition in (25), to a given pipeline "MC". Recall that the optimal redistribution function $G_{(V,H)}^{*}$ is a special case of $G_{(V,H)}^{\dagger}$. Now, using (25),

$$\int_{\mathcal{V}} \mathrm{d}v \int_{\mathcal{H}} \mathrm{d}h \left| F_{(V,H)}^{\text{MC-plus-ARCANE-}\dagger}(v,h) \right| = \int_{\mathcal{V}} \mathrm{d}v \int_{\mathcal{H}} \mathrm{d}h \left| F_{(V,H)}^{\mathrm{MC}}(v,h) + G_{(V,H)}^{\dagger}(v,h) \right| \tag{65}$$

$$= \int_{\mathcal{V}} \mathrm{d}v \left| \int_{\mathcal{H}} \mathrm{d}h \left( F_{(V,H)}^{\mathrm{MC}}(v,h) + G_{(V,H)}^{\dagger}(v,h) \right) \right| \tag{66}$$

$$= \int_{\mathcal{V}} \mathrm{d}v \left| F_{V}^{\mathrm{MC}}(v) \right|. \tag{67}$$

Plugging this into the definition of $\mathbb{SP}$ leads to

$$\mathbb{SP}_{\text{MC-plus-ARCANE-}\dagger}[V,H] = \mathbb{SP}_{\text{MC-plus-ARCANE-}\dagger}[V] \equiv \mathbb{SP}_{\mathrm{MC}}[V] \leq \mathbb{SP}_{\mathrm{MC}}[V,H]. \tag{68}$$

In this way, the ARCANE reweighting will lead to a reduction in the sign problem (assuming $\mathbb{SP}_{\mathrm{MC}}[V, H]$ is *strictly* greater than $\mathbb{SP}_{\mathrm{MC}}[V]$). The following conditional version of this equation can also be derived similarly:

$$\mathbb{SP}_{\text{MC-plus-ARCANE-}\dagger}\big[V,H \mid Y = y\big] = \mathbb{SP}_{\text{MC-plus-ARCANE-}\dagger}\big[V \mid Y = y\big] \tag{69}$$

$$\equiv \mathbb{SP}_{\mathrm{MC}}\big[V \mid Y = y\big] \leq \mathbb{SP}_{\mathrm{MC}}\big[V,H \mid Y = y\big]. \tag{70}$$

Note that ARCANE reweighting cannot reduce the sign problem below $\mathbb{SP}_{\mathrm{MC}}[V]$, since that is a fundamental limitation of modeling the quasi density $F_{V}^{\mathrm{MC}}$.

### 4.4 Discussions on Computational Benefits

**A simple discussion.** Let us consider the simple case where (a) the amortized cost per simulated event is roughly independent of choices made at the event generator level and (b) the total number of events required is directly related to $\mathbb{CV}_{\mathrm{MC}}^{2}[W]$. Now, performing ARCANE reweighting, with $G_{(V,H)}^{\dagger}$ as the redistribution function, followed by unweighting will lead to

$$\mathbb{CV}_{\text{MC-plus-ARCANE-}\dagger\text{-plus-unweighting}}^{2}[W] = \mathbb{SP}_{\mathrm{MC}}[V] \leq \mathbb{SP}_{\mathrm{MC}}[V,H] = \mathbb{CV}_{\text{MC-plus-unweighting}}^{2}[W]. \tag{71}$$

In the case where $\mathbb{SP}_{\mathrm{MC}}[V] = 0$ (i.e., the quasi density of $V$ is non-negative), this corresponds to a complete elimination of weight-variance. The total number of simulated events needed will be reduced accordingly, without an appreciable change to the amortized cost per event.

For completeness, Table 1 shows how the different additive components of $\mathbb{CV}_{\mathrm{MC}}^{2}[W]$ are affected by the two reweighting techniques. Each row represents a strictly non-negative component of $\mathbb{CV}_{\mathrm{MC}}^{2}[W]$. The complementarity of the two reweighting techniques is also evident from the table.

Table 1: The first column of this table lists four different non-negative terms, which add up to $\mathbb{CV}^2_{\text{MC}}[W]$. The middle and right columns describe how these components of $\mathbb{CV}^2_{\text{MC}}[W]$ are affected by ARCANE reweighting and rejection reweighting techniques. By performing ARCANE reweighting with $G_{(V,H)} = G^\dagger_{(V,H)}$ followed by unweighting, one can reduce $\mathbb{CV}^2[W]$ of the resulting sample down to $\mathbb{SP}_{\text{MC}}[W]$.

| Component of $\mathbb{CV}^2_{\text{MC}}[W]$ | How the component is affected by the reweighting technique? | |
| :---: | :---: | :---: |
| | **ARCANE reweighting** | **Rejection reweighting** |
| $\mathbb{SP}_{\text{MC}}[V]$ | Invariant | Invariant |
| $\mathbb{SP}_{\text{MC}}[V,H] - \mathbb{SP}_{\text{MC}}[V]$ | Can be fully eliminated (by setting $G_{(V,H)} \equiv G^\dagger_{(V,H)}$) | Invariant |
| $\dfrac{\mathbb{E}_{\text{MC}}\left[\text{Var}_{\text{MC}}\left[\lvert W\rvert \,\middle\vert\, V\right]\right]}{\mathbb{E}^2_{\text{MC}}[W]}$ | Can be fully eliminated (by setting $G_{(V,H)} \equiv G^*_{(V,H)}$) | Can be fully eliminated (by unweighting) |
| $\dfrac{\text{Var}_{\text{MC}}\left[\mathbb{E}_{\text{MC}}\left[\lvert W\rvert \,\middle\vert\, V\right]\right]}{\mathbb{E}^2_{\text{MC}}[W]}$ | Not invariant, but cannot be fully eliminated | Can be fully eliminated (by unweighting) |

**A more complex discussion.** Now let us consider the case where the amortized cost per event depends on the generator level efficiency of rejection reweighting as well as the costs associated with performing ARCANE reweighting. Furthermore, the sensitivity of the experimental analysis is not assumed to be directly related to $\mathbb{CV}^2[W]$. For these reasons, it is not assumed that complete unweighting of the events will be computationally feasible or even preferable in a given situation.

Let "MC-plus-rej-rewgtng" represent an existing Monte Carlo pipeline, which involves performing a base pipeline "MC" followed by an optional rejection reweighting step with $\alpha_{\text{func}}(V,H)$ as the acceptance probability function.[35] Choosing $\alpha_{\text{func}}(V,H) \equiv 1$ corresponds to not performing the rejection reweighting at all.

Now one can construct a modified event generation pipeline, which involves performing the pipeline MC followed by (i) ARCANE reweighting with the optimal redistribution function $G^*_{(V,H)}$ and (ii) a rejection reweighting step with a modified acceptance probability given by

$$\alpha'_{\text{func}}(V,H) \equiv \mathbb{E}_{\text{MC}}\left[\alpha_{\text{func}}(V,H) \,\middle\vert\, V\right]. \tag{72}$$

By construction, $P^{\text{MC-plus-rej-rewgtng}}_V \equiv P^{\text{MC-plus-ARCANE-*-plus-rej-rewgtng}}_V$, since the average acceptance rate for each value of $V$ is the same for the rejection reweighting step in both pipelines. Since incorporating ARCANE reweighting does not change the quasi density of $V$, this also implies that

$$\mathbb{E}_{\text{MC-plus-ARCANE-*-plus-rej-rewgtng}}\left[W \,\middle\vert\, V = v\right] \equiv \mathbb{E}_{\text{MC-plus-rej-rewgtng}}\left[W \,\middle\vert\, V = v\right]. \tag{73}$$

Furthermore, since (i) the ARCANE reweighting step (with the optimal redistribution function) completely eliminates the weight-variance among events with the same value of $V$, and (ii) the modified acceptance rate function $\alpha'_{\text{func}}(V,H)$ is independent of $H$, it follows that, for all $v \in \mathcal{V}$,

$$\text{Var}_{\text{MC-plus-ARCANE-*-plus-rej-rewgtng}}\left[W \,\middle\vert\, V = v\right] = 0 \tag{74}$$

---

[35] Since event weights are deterministic functions of $(V,H)$, acceptance probability functions based on the event-weight can be written as functions of $(V,H)$.

$$\implies \quad \mathbb{CV}^2_{\text{MC-plus-ARCANE-}*\text{-plus-rej-rewgtng}}\Big[W \mid V = v\Big] \leq \mathbb{CV}^2_{\text{MC-plus-rej-rewgtng}}\Big[W \mid V = v\Big]. \quad (75)$$

From this and Statement 1, it follows that the number of simulated events needed under "MC-plus-ARCANE-$*$-plus-rej-rewgtng" will be less than or equal to the number of events needed under MC-plus-rej-rewgtng, at least for histogram-based analyses. In other words, roughly speaking, there is a non-negative computational benefit to incorporating ARCANE reweighting into an event-generation pipeline, regardless of details like rejection reweighting efficiency, preferential biasing in event generation [2, 59], etc. Whether this benefit is worth the various costs associated ARCANE reweighting will depend on the specific use-case, as well as how close the implementation of ARCANE is to the optimal one with $G_{(V,H)} = G^*_{(V,H)}$.

Note that the modified acceptance probability function $\alpha'_{\text{func}}$ may not be the best choice of acceptance probability function, after performing ARCANE reweighting. It is used here simply to construct a pipeline MC-plus-ARCANE-$*$-plus-rej-rewgtng that is guaranteed to be at least as good as MC-plus-rej-rewgtng, as per (75). However, one will likely be able to get better results with a different acceptance function.

## 4.5 Effect of ARCANE Reweighting on Alternative Measures of Weight-Variability

Some alternative measures of weight-*variability* may be more relevant in certain situations than the weight-variance. For example, unweighting efficiency of a Monte Carlo pipeline depends more on the maximum absolute event weight (pre-unweighting) than on weight-variance. In addition to the weight-variance, ARCANE reweighting can reduce several other measures of variability of event-weights; some of these are listed next. Using Jensen's inequality and the law of total expectation, it can be shown that

$$\mathbb{E}_{\text{MC-plus-ARCANE-}*}\Big[f_{\text{convex}}(W) \mid X(V) = x\Big] \leq \mathbb{E}_{\text{MC}}\Big[f_{\text{convex}}(W) \mid X(V) = x\Big], \quad (76a)$$

for any convex function $f_{\text{convex}} : \mathbb{R} \longrightarrow \mathbb{R}$, any event attribute $X$ that is a (possibly random) function of $V$, and any possible value $x$ for $X$. Choosing $f_{\text{convex}}(w)$ to be $w^2$ leads to the corresponding inequality for the weight-variance. From the fact that the arithmetic mean of a population will be bounded by its extreme values, it can be seen that

$$\mathbb{ESS\,SUP}_{\text{MC-plus-ARCANE-}*}\Big[W \mid X(V) = x\Big] \leq \mathbb{ESS\,SUP}_{\text{MC}}\Big[W \mid X(V) = x\Big], \quad (76b)$$

$$-\mathbb{ESS\,INF}_{\text{MC-plus-ARCANE-}*}\Big[W \mid X(V) = x\Big] \leq -\mathbb{ESS\,INF}_{\text{MC}}\Big[W \mid X(V) = x\Big], \quad (76c)$$

where $\mathbb{ESS\,SUP}$ and $\mathbb{ESS\,INF}$ represent the essential supremum and essential infimum, respectively, under the relevant probability measure corresponding to the subscript "MC$\cdots$". These can be thought of as measure-theoretic analogues of maximum and minimum, respectively.

Similar inequalities can also be derived for the magnitudes of event-weights. For example,

$$\mathbb{E}_{\text{MC-plus-ARCANE-}*}\Big[f_{(\text{convex,inc})}\big(|W|\big) \mid X(V) = x\Big] \leq \mathbb{E}_{\text{MC}}\Big[f_{(\text{convex,inc})}\big(|W|\big) \mid X(V) = x\Big], \quad (76d)$$

for any convex, increasing, real-valued function $f_{(\text{convex,inc})}$ defined on non-negative reals. This follows from the fact that the function $f$, defined on reals as $f(w) \equiv f_{(\text{convex,inc})}(|w|)$ is convex. As a special case of the previous equation, one can show that

$$\left(\mathbb{E}_{\text{MC-plus-ARCANE-}*}\Big[|W|^p \mid X(V) = x\Big]\right)^{1/p} \leq \left(\mathbb{E}_{\text{MC}}\Big[|W|^p \mid X(V) = x\Big]\right)^{1/p}, \quad \forall p \geq 1. \quad (76e)$$

This inequality covers the effect of optimal ARCANE reweighting on the (conditional) generalized means or the power means of $|W|$, with $p \geq 1$. The $p = 1$ case of this inequality

corresponds to the fact that optimal ARCANE reweighting cannot worsen the sign problem. Taking the large $p$ limit of this inequality leads to

$$\mathbb{ESS\,SUP}_{\text{MC-plus-ARCANE-}*}\Big[|W|\ \Big|\ X(V)=x\Big]\ \leq\ \mathbb{ESS\,SUP}_{\text{MC}}\Big[|W|\ \Big|\ X(V)=x\Big]. \qquad (76f)$$

Given that $\mathbb{E}_{\text{MC-plus-ARCANE-}*}[W\mid X(V)=x]=\mathbb{E}_{\text{MC}}[W\mid X(V)=x]$ by construction, each inequality in (76) represents a reduction of a measure of weight-variance via ARCANE reweighting. Choosing $X(V)$ to be a constant leads to unconditional variants of each inequality in (76).

**Effect on unweighting efficiency.** The unconditional variant of (76f) roughly states that the maximum value of $|W|$ under "MC-plus-ARCANE-$*$" will be less than or equal to that under MC. In the situation where the weights under MC are non-negative to being with, this would imply that the unweighting efficiency cannot be worsened by applying the optimal ARCANE reweighting. However, for the general case where weights can be positive or negative, applying ARCANE reweighting can potentially worsen the unweighting efficiency.[36] However, such a situation should not be interpreted as worsening of performance by the incorporation of ARCANE reweighting. Recall, from the discussion in Section 4.4, that optimal ARCANE reweighting will bring a non-negative **benefit** to the performance (not counting the **costs** associated with its implementation).

To understand this, consider the following toy example: Let $V$ be a discrete event-attribute that is either 0 or 1. Under a given event generation pipeline MC, let the event-weights be of constant magnitude, say 1. Furthermore, let the probabilities of $(V,W)$ being $(0,+1)$, $(0,-1)$, $(1,+1)$, and $(1,-1)$ be 0.3, 0.2, 0.4, and 0.1, respectively, under MC. Now, if one incorporates the optimal ARCANE reweighting procedure, $(V,W)$ would be $(0,0.2)$ with probability 0.5 and $(1,0.6)$ with probability 0.5. In this example, under the original pipeline MC, the unweighting efficiency can be as high as 1, since the weights have a constant magnitude. On the other hand, the unweighting efficiency under MC-plus-ARCANE-$*$ can be at most 2/3. However, after unweighting, the weights will be equal under MC-plus-ARCANE-$*$-plus-unweighting, which is not the case for MC-plus-unweighting. So, a comparison based only on unweighting efficiencies would not be fair. In this example, even without any rejection reweighting (i.e., with a rejection reweighting efficiency of 1), the weight distribution with ARCANE reweighting is better than the one under MC-plus-unweighting. This is in line with the arguments in Section 4.4, based on the modified acceptance probability function $\alpha'_{\text{func}}$ in (72).

## 4.6 Effect of Subsequent Processing of the Event

Consider a situation where the subsequent processing of the weighted event $(V,W)$, in the rest of the event production pipeline, can lead to further multiplicative weight updates. Let $(V_{\text{final}},W_{\text{final}})$ be the final weighted visible event, with $W_{\text{final}}\equiv W\times W_{\text{update}}$. Let

$$P^{\text{MC}}_{(V_{\text{final}},W_{\text{update}}\mid V,W)}(v_{\text{final}},w_{\text{update}}\mid v,w)$$

be independent of $w$. It can be shown that the inequalities in (76) will continue to hold even if $V$ and $W$ are replaced with $V_{\text{final}}$ and $W_{\text{final}}$, respectively (with "ARCANE-$*$" in the inequalities still representing optimal ARCANE reweighting at the original stopping point). In other words, the optimality of $G^*_{(V,H)}$ holds irrespective of how the event will be processed subsequently, provided there are no weight-dependent procedures like unweighting in the

---

[36]This is because both $\mathbb{E}[|W|]$ and $\mathbb{ESS\,SUP}[|W|]$ can be lowered by ARCANE reweighting; their ratio will be the unweighting efficiency, assuming that one uses $\mathbb{ESS\,SUP}[|W|]$ as the reference weight in (31).

subsequent processing. One way to understand this is as follows. The variance of $W$ can be decomposed as

$$\text{Var}_{\text{MC}}[W] = \underbrace{\mathbb{E}_{\text{MC}}\Big[\text{Var}_{\text{MC}}[W \mid V]\Big]}_{\text{This weight-variability always bad}} + \underbrace{\text{Var}_{\text{MC}}\Big[\mathbb{E}_{\text{MC}}[W \mid V]\Big]}_{\text{This weight-variability could be good}} . \quad (77)$$

The weight-variability captured by the second term (on the right-hand-size) could be beneficial, depending on context (like with intentional biasing [2, 59]). However, as a rule of thumb, the weight-variablity captured by the first term always has a detrimental effect on performance,[37] as long as the subsequent processing of the event depends only on $V$. ARCANE reweighting, by construction, does not affect the variability captured by the second term of (77). Furthermore, optimal ARCANE reweighting completely eliminates the first term of (77). This is roughly the rationale behind the overarching usefulness of ARCANE reweighting.

**Situations where $F_V^{\text{MC}}(v)$ itself can be negative.** In some situations the quasi density of the visible event attributes, namely $F_V^{\text{MC}}(v)$ itself may be negative for some values of $v$. This may be acceptable for various reasons. If $V$ is not an experimentally observable event-attribute and the map from $V$ to the observable attributes is probabilistic and many-to-many, the subsequent processing of the event $V$ could still lead to non-negative quasi densities for observable attributes.[38] Furthermore, if one applies ARCANE reweighting within a *subset* of the latent generator pathways (cf. Section 5.1) or for an intermediate sequence of generation steps (cf. Section 5.3), then the corresponding target quasi density may take negative values.

In such situations, optimal ARCANE reweighting may actually *increase* the value of $f^{(\text{neg})}$ and/or $f^{(\text{neg,local})}(V = v)$ away from 1/2, for the event generation steps or stages under consideration. However, this still indicates a reduction in the sign problem. Assuming that the final observable event-attributes have a non-negative quasi density, optimal ARCANE redistribution within any sequence of steps can only lead to a reduction in the overall negative weights fraction.

For the two-step procedure to sample $(V_{\text{final}}, W_{\text{final}})$ described above, if the quasi density $F_V^{\text{MC}}$ is not non-negative and $\text{sgn}(W_{\text{update}})$ is guaranteed to match the sign of $F_V^{\text{MC}}(V)$, then one can solve the overall sign problem in the generation of $V_{\text{final}}$ simply by performing ARCANE reweighting at the chosen stopping point (where $V$ is the "visible" event-attribute). However, in general, if the quasi density $F_V^{\text{MC}}$ is not non-negative, one may have to choose a different stopping point within the generator to perform ARCANE reweighting at, in order to solve the overall sign problem.

# 5 Some Generalizations of ARCANE Reweighting

There are a few straightforward generalizations of the basic ARCANE reweighting technique, which will be discussed next. These could be useful in tackling different use-cases.

## 5.1 Flexibility in the Choice of Visible and Hidden Attributes

The ARCANE redistribution technique has so far been discussed in terms of $V$, which represents all and only the visible event-attributes, and $H$, which represents all and only the invisible event-attributes. However, it can be extended as follows. First, $V$ can be replaced with the

---

[37]This is true, except in some applications like Monte Carlo integration with stratified sampling [61].

[38]As a moot point, it is also possible that the physics model represented by $F_V^{\text{MC}}$ actually leads to negative values for the quasi densities of observable event-attributes; our models describe nature only approximately, so may be unphysical to an acceptably small degree.

meta-attribute $S \in \mathcal{S}$, which includes all the visible event-attributes and possibly some hidden event-attributes as well. $S$ can be thought of as information that is **sufficient** for further processing each event in the simulation and analysis pipeline; $V$ will be fully determined by $S$. Secondly, $H$ can be replaced with a meta-attribute $L$, which represents a *subset* of the information that is non-contained in $S$. In other words, $(S, L)$ need not cover all sources of randomness in the event generation. However, one still needs to be able to compute $P_{(S,L)}^{\text{MC}}$ to perform the reweighting. For concreteness, analogous to (17), the reweighting will be performed as follows:

$$W_{\text{ARCANE}} = W_{\text{MC}} + W_{\Delta}\,, \qquad \text{where} \quad W_{\Delta} \equiv \frac{G_{(S,L)}(S,L)}{P_{(S,L)}^{\text{MC}}(S,L)}\,. \tag{78}$$

Such a reweighting, with appropriate conditions on $G_{(S,L)}$, will preserve the quasi density of $S$, consequently preserving the quasi density of $V$ as well. If $W_{\text{MC}}$ is not exactly determined by $(S, L)$, then this reweighting will not be able to make $W_{\text{ARCANE}}$ exactly match $\mathbb{E}_{\text{MC}}[W_{\text{MC}} | S]$ for each event; this may be acceptable depending on the use-case at hand.

The leeway in choosing $S$ could be useful, for example, when there are many discrete channels for producing an event with a given value of $V$. In that case, instead of redistributing the contributions among all the channels, one could create a few groups of channels and only redistribute the contributions within each group—this might be sufficient for the purpose of reducing the negative weights problem. The corresponding meta event-attribute $S$ (whose quasi density is preserved) would contain $V$ as well as the label of the channel-group the event was sampled from.

## 5.2   Multiple Reweighting Terms

Instead of having a single additive reweighting term, one can extend ARCANE reweighting to use multiple reweighting terms as follows:

$$W_{\text{ARCANE}} = W_{\text{MC}} + \sum_{i=1}^{n} W_{\Delta,i}\,, \qquad \text{where} \qquad W_{\Delta,i} \equiv \frac{G_{(S,L_i)}^{(i)}(S,L_i)}{P_{(S,L_i)}^{\text{MC}}(S,L_i)}\,, \tag{79}$$

where each $L_i$ could be a different subset of the hidden event-attributes. The quasi density of $S$ will be preserved by such a weighting if proper coverage is ensured and the following condition is satisfied:

$$\sum_{i=1}^{n} G_S^{(i)}(s) \equiv 0\,, \qquad \forall s \in \mathcal{S}\,, \tag{80}$$

where $G_S^{(i)}$ is the marginal quasi density corresponding to $G_{(S,L_i)}^{(i)}$ in (79).

## 5.3   Application to an Intermediate Sequence of Generation Steps

So far, ARCANE reweighting has been discussed as a way to alter the weight-distribution of events produced by a sequence of steps from the *start* until some arbitrary point in the event generation pipeline. However, the technique can also be applied to an *intermediate* sequence of steps in the generation pipeline. Consider a two-stage event generation setup to produce $(A, V, H)$ whose quasi density is given by

$$F_{(A,V,H)}^{\text{MC-overall}}(a,v,h) = F_A^{\text{MC-prelude}}(a) \, \times \, F_{(V,H;A)}^{\text{MC}}(v,h\,;\,a)\,. \tag{81}$$

A preliminary generation stage produces $A$ with a quasi density $F_A^{\text{MC-prelude}}$. Then, $(V, H)$ is produced with a parameterized quasi density $F_{(V,H\,;A)}^{\text{MC}}$; let this be the "main" MC stage under consideration. $V$ and $H$ have the same meanings as before, in regard to their visibility to the subsequent simulation stages. Let $W_{\text{prelude}}$ be the event weight after the preliminary generation stage and let $W_{\text{MC}}$ be the weight update corresponding to the main MC stage. The overall weight of the event is given by $W_{\text{overall}} = W_{\text{prelude}} \times W_{\text{MC}}$. Now, one can reweight each event as follows:

$$W_{\text{overall,ARCANE}} = W_{\text{prelude}} \times \left( W_{\text{MC}} + W_{\Delta} \right), \qquad \text{where} \qquad W_{\Delta} \equiv \frac{G_{(V,H;A)}(v, h \; ; \; a)}{P_{(V,H;A)}^{\text{MC}}(v, h \; ; \; a)}. \tag{82}$$

As before, the quasi density of $(A, H)$ will be preserved by such a reweighting if $G_{(V,H;A)}$ integrates with respect to $H$ to 0 (and satisfies the necessary conditions to ensure proper coverage). In this way, one can apply ARCANE reweighting to an intermediate sequence of the MC pipeline. Furthermore, by setting $G_{(V,H;A)}(v, h \; ; \; a)$ to be exactly zero for some values of $a$, one can apply ARCANE reweighting only within certain branches of the MC-generation-pipeline. Note that in order to perform the reweighting in (82), one needs access to $W_{\text{prelude}}$ and $W_{\text{MC}}$ separately, and not just the combined $W_{\text{overall}}$.

As an example application, one can use this technique with the weighted veto algorithm [8–12] to reduce the weight-variance among events that go through different sequences of rejected emission proposals, but are otherwise identical. Here the weight redistribution will be performed only within parton shower stage, ignoring the MC history of the event leading up to the parton showering.

## 5.4 ARCANE Aware Sampling; Making History

As discussed in Section 3.3, ARCANE reweighting cannot, *by itself*, change the sampling probability density $P_V^{\text{MC}}$ of the visible attributes. However, one can use the technique to modify the sampling density as follows. First, one can modify an existing generation pipeline by adding a new pathway (i.e., by "making new MC histories") for creating events:

$$P_V^{\text{MC-plus-new-history}}(v) = P_{(V,Z)}^{\text{MC-plus-new-history}}(v, 0) + \int_{\mathcal{H}} \mathrm{d}h \; P_{(V,Z,H)}^{\text{MC-plus-new-history}}(v, 1, h), \tag{83}$$

$$\text{where} \qquad P_{(V,Z)}^{\text{MC-plus-new-history}}(v, 0) = \zeta \; P_V^{\text{new-pathway}}(v), \tag{84}$$

$$\text{and} \qquad P_{(V,Z,H)}^{\text{MC-plus-new-history}}(v, 1, h) = (1 - \zeta) \, P_{(V,H)}^{\text{MC}}(v, h). \tag{85}$$

Here $Z \in \{0, 1\}$ is the latent variable indicating whether an event was sampled via the original generator (indicated by $Z = 1$) or the new pathway (indicated by $Z = 0$). $\zeta \in [0, 1)$ is the probability of choosing the new pathway, and $(1 - \zeta)$ is the probability of proceeding to the original generation pipeline "MC". For events sampled from the new pathway, the event-weights will be set to zero. For events sampled via the original generator, the event weight $W_{\text{MC}}$ will be replaced by

$$W_{\text{MC-plus-new-history}}(V, Z = 1, H) = \frac{W_{\text{MC}}(V, H)}{\zeta}. \tag{86}$$

In this way, one can ensure that $F_V^{\text{MC-plus-new-history}} \equiv F_V^{\text{MC}}$. Now, the contributions of the different latent pathways, identified by $Z$ and (possibly) $H$, that lead to the same $V$ can be redistributed via ARCANE reweighting.

A use-case for this technique is the following. Typically, the different importance sampling steps in an FCMC pipeline are optimized locally, to provide a good performance (from a weight-variance or an unweighting-efficiency perspective) for that particular step. If each step of the

FCMC pipeline is implemented optimally, then the implementation of the overall pipeline will also be optimal. However, if one incorporates ARCANE reweighting, then locally optimizing the importance sampling steps may not lead to a good sampling distribution for the modified quasi density $F_{(V,H)}^{\text{ARCANE}}$. This will be true especially when different regions in the phase space of $V$ have different degrees of the sign problem. This raises the possibility for "ARCANE aware sampling", i.e., choosing the sampling distribution, accounting for the fact that the events will undergo ARCANE reweighting. Creating a new generator pathway, as described above is one way to modify the sampling distribution to perform well (again, from a weight-variance or an unweighting-efficiency perspective) under ARCANE reweighting. Machine learning methods could be useful in learning an appropriate distribution $P_V^{\text{new-pathway}}$ for this purpose.

## 5.5 Applications Beyond Weight Redistribution

So far, ARCANE reweighting has been discussed as a way to redistribute the contributions of different pathways using a redistribution function $G_{(V,H)}$. However, it can be extended beyond such applications as follows. Consider a quasi density $F_X^{\text{combined}}$ of a complicated, multi-dimensional event-attribute $X \in \mathcal{X}$ given by

$$F_X^{\text{combined}}(x) \equiv F_X^{(1\text{-FCMC-able})}(x) + F_X^{(2)}(x), \qquad \forall x \in \mathcal{X}, \qquad (87)$$

where $F_X^{(1\text{-FCMC-able})}$ is available in a straightforwardly FCMC-able form, and $F_X^{(2)}(x)$ can be computed for specific values of $x$. Now, one can sample events from $F_X^{\text{combined}}$ by sampling events from $F_X^{(1\text{-FCMC-able})}$ and incorporating the contribution of $F_X^{(2)}$ via an additive reweighting, with $W_\Delta = F_X^{(2)}(X)/P_X^{\text{MC}}(X)$.

Furthermore, consider a situation where $F^{(1\text{-FCMC-able})}$ and $F^{(2)}$ are defined over different sets of latent attributes, e.g., as follows:

$$
\begin{aligned}
F_V^{\text{combined}}(v) \equiv &\int_{\mathcal{Z}_0} dz_0 \int_{\mathcal{Z}_1} dz_1 \, F_{(V,Z_0,Z_1)}^{(1\text{-FCMC-able})}(v, z_0, z_1) \\
&+ \int_{\mathcal{Z}_0} dz_0 \int_{\mathcal{Z}_2} dz_2 \, F_{(V,Z_0,Z_2)}^{(2)}(v, z_0, z_2), \qquad \forall v \in \mathcal{V},
\end{aligned}
\qquad (88)
$$

where $Z_0$ contains latent attributes shared by both terms, while $Z_1$ and $Z_2$ could contain latent attributes specific only to the first and the second term, respectively. Now, one can make the two terms on the right-hand-side be defined over the same phase space as follows:

$$
\begin{aligned}
F_{(V,Z_0,Z_1,Z_2)}^{\text{combined}}(v, z_0, z_1, z_2) \equiv\ & F_{(V,Z_0,Z_1)}^{(1\text{-FCMC-able})}(v, z_0, z_1) \quad F_{(Z_2;V,Z_0,Z_1)}^{(\text{unit-norm-1})}(z_2 \, ; \, v, z_0, z_1) \\
&+ F_{(V,Z_0,Z_2)}^{(2)}(v, z_0, z_2) \quad F_{(Z_1;V,Z_0,Z_2)}^{(\text{unit-norm-2})}(z_1 \, ; \, v, z_0, z_2),
\end{aligned}
\qquad (89)
$$

where $F_{(Z_2;V,Z_0,Z_1)}^{(\text{unit-norm-1})}$ and $F_{(Z_1;V,Z_0,Z_2)}^{(\text{unit-norm-2})}$ are arbitrary, unit-normalized (parameterized) quasi densities of $Z_2$ and $Z_1$, respectively. Now both terms have the same set of arguments, namely $V$, $Z_0$, $Z_1$, and $Z_2$, so one can sample events from the first term and incorporate the contribution of the second via an additive reweighting. This may be reminiscent of the Born spreading technique [44], with the added intricacy of a deferred reweighting.

The label "ARCANE reweighting" is intended to apply to any such approach where some contribution (to the quasi density of MC datasamples) is incorporated via a deferred, additive reweighting. The generalization discussed here provides an alternative approach to reduce the negative weights problem in MC@NLO event generation, where one can just sample $\mathbb{S}$-type events and incorporate the $\mathbb{H}$-type contribution via ARCANE reweighting. This alternative approach, dubbed "hard remainder spreading" is also discussed in the companion paper Ref. [7].

The ability to incorporate contributions in this way, after the complete visible event *V* has been generated, can potentially lead to new theoretical/phenomenological formalisms, e.g., for interfacing higher order matrix-element calculations with parton showers, without the constraint of being straightforwardly FCMC-able.

# 6 Summary, Conclusions, and Outlook

This paper has introduced a new Monte Carlo technique called ARCANE reweighting to tackle the negative weights problem encountered in collider event generation. The technique has been explained in the context of a generic event generation task in this paper. It can be applied to specific situations where negative weights are encountered, like event generation using the MC@NLO formalism. ARCANE reweighting works carefully modifying the contributions of different pathways within an event production pipeline that leads the same final event, in such a way that the overall weighted distribution of the visible events remains unaffected. This is performed using an additive correction to the event-weight that depends on the Monte Carlo history of the event. This additive reweighting term averages to zero, over the different generator pathways leading to the same final event. The technique is exact and does not introduce any biases in the Monte Carlo predictions or any correlations between events. By computing the additive weight correction after the entire event is generated, ARCANE sidesteps the Forward Chain Monte Carlo paradigm, as described in Section 2.1, which the existing approaches to event generation fall under. The ability to perform such a deferred reweighting can potentially facilitate new theory formalisms, e.g., for interfacing NLO matrix-element calculations with parton showers; exploring this possibility is beyond the scope of this work.

An important aspect of implementing the ARCANE reweighting technique is the construction of the ARCANE redistribution function, which determines the amount by which the contributions of the different generator pathways are increased or decreased. The success of the ARCANE reweighting technique relies crucially on the quality of the redistribution function used. The companion paper, Ref. [7], (i) outlines a construction of a good redistribution function and (ii) demonstrates a successful application of ARCANE reweighting for the generation of $(e^+e^- \longrightarrow q\bar{q} + 1\,\texttt{jet})$ events using the MC@NLO formalism. Ref. [7] also discusses how the technique can be extended to other situations of interest, including hadronic collisions, in a systematic manner.

The implementation of ARCANE reweighting cannot be decoupled from the implementation of event generators. Tracking and recording the hidden attributes that identify the different latent generator pathways is an essential aspect of ARCANE. Likewise, computing the sampling probabilities of any given pathway in the generator is also a requisite. Furthermore, knowledge of the internal workings of the event generator can help with engineering a good redistribution function (instead of learning it from Monte Carlo datasets). As a sidenote, in regions of the phase space of *V* where the local sign problem is severe, sufficient care should be taken in the implementation of the generators to avoid catastrophic cancellations; this is true regardless of whether or not ARCANE reweighting is incorporated in the event generator.

The incorporation of ARCANE reweighting into an event generation pipeline, *for the most part*, does not require any changes to how the events will be used in an experimental analysis. There could be a few situations where ARCANE-reweighted-events require a different treatment. For example, when reweighting a dataset to make it correspond to a different set of values for event-generator parameters [50–56], one must apply the multiplicative reweighting factor(s) only to the original weight of each event, i.e., $W_{\mathrm{MC}}$ in (17). The ARCANE correction term $W_\Delta$ in (17) can be left as is. Note that a nearly optimal redistribution function for one choice of values for the generator parameters, represented by $\boldsymbol{\theta}$, may not be close to optimal

for a different choice. To rectify this, one could use a redistribution function $G_{(V,H)}(V, H ; \boldsymbol{\theta})$, that is conditioned on $\boldsymbol{\theta}$, constructed to perform well over a range of values for $\boldsymbol{\theta}$. Here $\boldsymbol{\theta}$ can include the theoretical/phenomenological parameters characterizing the quasi density $F_{(V,H)}^{\text{MC}}$ as well as logistical parameters characterizing the sampling probability density $P_{(V,H)}^{\text{MC}}$.

As an additional contribution, this paper raises, in Section 1.3.2 and Appendix A, an issue with positive resampling and some other related techniques [34–38, 40, 41] for reducing the negative weights problem, namely that such techniques lead to dependent event samples.

# Acknowledgements

The author thanks Daniel Hackett, Stefan Höche, Kyoungchul Kong, Nicholas Manganelli, Konstantin Matchev, Kirtimaan Mohan, Gabriel Perdue, and in particular, Stephen Mrenna, Kevin Pedro, Nicholas Smith, and Manuel Szewc for useful discussions and/or feedback on this manuscript. The author thanks the Aspen Center for Physics for hospitality and support during May–June of 2022 and 2023.

**Funding information**   This document was prepared using the resources of the Fermi National Accelerator Laboratory (Fermilab), a U.S. Department of Energy, Office of Science, Office of High Energy Physics HEP User Facility. Fermilab is managed by Fermi Forward Discovery Group, LLC, acting under Contract No. 89243024CSC000002. The author is supported by the U.S. Department of Energy, Office of Science, Office of High Energy Physics QuantISED program under the grants "HEP Machine Learning and Optimization Go Quantum", Award Number 0000240323, and "DOE QuantiSED Consortium QCCFP-QMLQCF", Award Number DE-SC0019219. This work was partially performed at the Aspen Center for Physics, which is supported by National Science Foundation grant PHY-1607611.

# A   Non IID-ness in Positive Resampling and Related Techniques

Consider a binned analysis performed using $N \geq 2$ weighted, *exchangeable* datapoints $\left\{(W_i, B_i)\right\}_{i=1}^N$, where $B_i$ is the bin index of the $i$-th datapoint. The normalized histogram height for a given bin $b$ of the histogram, which is an estimate for the quasi density of that bin, is given by

$$\widehat{F}_B(b) = \frac{1}{N} \sum_{i=1}^{N} \Big[ W_i\, \delta_{\text{K}}(B_i, b) \Big], \qquad \text{where } \ \delta_{\text{K}}(b, b') \equiv \begin{cases} 1, & \text{if } b = b', \\ 0, & \text{otherwise.} \end{cases} \tag{A.1}$$

**Some results for exchangeable datapoints.**   Using Bienaymé's identity, the exchangeability of datapoints, and the laws of total variance and total covariance, it can be shown that the variance of $\widehat{F}_B(b)$ is given by

$$\text{Var}\big[\widehat{F}_B(b)\big] = \frac{1}{N}\,\text{Var}\Big[W_1\,\delta_{\text{K}}(B_1, b)\Big] + \underbrace{\frac{N-1}{N}\,\text{Cov}\Big[W_1\,\delta_{\text{K}}(B_1, b) \ , \ W_2\,\delta_{\text{K}}(B_2, b)\Big]}_{= 0 \text{ if datapoints are IID}} \tag{A.2}$$

$$
\begin{aligned}
&= \frac{1}{N}\, P_{B_1}(b)\, \mathrm{Var}\big[W_1 \,\big|\, B_1 = b\big] + \frac{1}{N}\, P_{B_1}(b)\left(1 - P_{B_1}(b)\right)\mathbb{E}^2\big[W_1 \,\big|\, B_1 = b\big] \\
&\quad + \underbrace{\frac{N-1}{N}\, P_{(B_1,B_2)}(b,b)\, \mathrm{Cov}\big[W_1, W_2 \,\big|\, B_1 = B_2 = b\big]}_{=\,0 \text{ if datapoints are IID}} \\
&\quad + \underbrace{\frac{N-1}{N}\, P_{(B_1,B_2)}(b,b)\, \mathbb{E}^2\big[W_1 \,\big|\, B_1 = B_2 = b\big] - \frac{N-1}{N}\, P_{B_1}^2(b)\, \mathbb{E}^2\big[W_1 \,\big|\, B_1 = b\big]}_{=\,0 \text{ if datapoints are IID}}.
\end{aligned} \tag{A.3}
$$

$$
\begin{aligned}
&= \frac{1}{N}\, P_{B_1}(b)\, \mathbb{E}\big[W_1^2 \,\big|\, B_1 = b\big] - \frac{1}{N}\, P_{B_1}^2(b)\, \mathbb{E}^2\big[W_1 \,\big|\, B_1 = b\big] \\
&\quad + \underbrace{\frac{N-1}{N}\, P_{(B_1,B_2)}(b,b)\, \mathbb{E}\big[W_1 W_2 \,\big|\, B_1 = B_2 = b\big] - \frac{N-1}{N}\, P_{B_1}^2(b)\, \mathbb{E}^2\big[W_1 \,\big|\, B_1 = b\big]}_{=\,0 \text{ if datapoints are IID}}.
\end{aligned} \tag{A.4}
$$

Likewise, it can be shown that the covariance between $\widehat{F}_B(b)$ and $\widehat{F}_B(b')$ for $b \neq b'$ is given by

$$
\begin{aligned}
\mathrm{Cov}\big[\widehat{F}_B(b)\,,\,\widehat{F}_B(b')\big] &= \frac{1}{N}\,\mathrm{Cov}\Big[W_1\,\delta_{\mathrm{K}}(B_1,b)\,,\,W_1\,\delta_{\mathrm{K}}(B_1,b')\Big] \\
&\quad + \underbrace{\frac{N-1}{N}\,\mathrm{Cov}\Big[W_1\,\delta_{\mathrm{K}}(B_1,b)\,,\,\,W_2\,\delta_{\mathrm{K}}(B_2,b')\Big]}_{=\,0 \text{ if datapoints are IID}}
\end{aligned} \tag{A.5}
$$

$$
\left.
\begin{aligned}
&= -\frac{1}{N}\, P_{B_1}(b)\, P_{B_2}(b')\, \mathbb{E}\big[W_1 \,\big|\, B_1 = b\big]\,\mathbb{E}\big[W_2 \,\big|\, B_1 = b'\big] \\
&\quad + \frac{N-1}{N}\, P_{(B_1,B_2)}(b,b')\, \mathbb{E}\big[W_1 W_2 \,\big|\, B_1 = b, B_2 = b'\big] \\
&\quad - \frac{N-1}{N}\, P_{B_1}(b)\, P_{B_2}(b')\, \mathbb{E}\big[W_1 \,\big|\, B_1 = b\big]\,\mathbb{E}\big[W_2 \,\big|\, B_2 = b'\big].
\end{aligned}
\right\}\; = 0 \text{ if datapoints are IID} \tag{A.6}
$$

For the case of non-IID (but exchangeable) datapoints, one could use these equations to construct error *estimation* formulas, which rely on multiple datasubsets that are identically distributed and mutually independent of each other (the datapoints within a given subset could be mutually dependent).

**Simplification for IID datapoints.**   Let us additionally assume that the datapoints are IID. In this case, an estimate for the statistical uncertainty in $\widehat{F}_B(b)$ can be computed, e.g., as[39]

$$
\widehat{\sigma}_B(b) = \frac{1}{\sqrt{N}}\sqrt{\frac{1}{N}\sum_{i=1}^{N}\Big[W_i^2\,\delta_{\mathrm{K}}(B_i,b)\Big]}\,. \tag{A.7}
$$

Furthermore, in the case of IID datapoints, for two bins $b \neq b'$ with $P_{B_1}(b), P_{B_1}(b') \ll 1$, $\widehat{F}_B(b)$ and $\widehat{F}_B(b')$ will be approximately[40] uncorrelated.

---

[39]If the probability of an event being in the bin $b$ is not small, then this formula will significantly overestimate the error. In such cases, one could use the sample standard deviation of $[W_i\,\delta_{\mathrm{K}}(B_i,b)]$ divided by $\sqrt{N}$ as the error estimate.

[40]If the total number of events $N$ is not a constant, but Poisson distributed (and the distribution of individual datapoints is independent of $N$, as is typical), then $\widehat{F}_B(b)$ and $\widehat{F}_B(b')$ will be exactly independent.

**Data dependency issue illustrated using bin-based resampling.** Let us redistribute the weights of events evenly within each bin as follows:

$$W_i' \equiv \frac{\sum\limits_{j=1}^{N}\left[W_j\,\delta_{\mathrm{K}}(B_j,B_i)\right]}{\sum\limits_{j=1}^{N}\delta_{\mathrm{K}}(B_j,B_i)}\,. \tag{A.8}$$

Now, one can compute a new histogram height, say $\widehat{F}_B'(b)$, and a corresponding new error estimate, say $\widehat{\sigma}_B'(b)$, using the formulas in (A.1) and (A.7), with the modified weights $W_i'$ in place of $W_i$. It is easy to see that $\widehat{F}_B'(b)$ will be exactly the same as $\widehat{F}_B(b)$. However, the error estimate $\widehat{\sigma}_B'(b)$ would be lesser than $\widehat{\sigma}_B(b)$, unless the original weights in bin $b$ were all equal to being with. This indicates an underestimation of the error in $\widehat{F}_B'(b)$ by $\widehat{\sigma}_B'(b)$, which is problematic[41]; this was also noted in Ref. [35]. This discussion applies even for the alternative weight redistribution of the form in Refs. [34, 36]; there the weights are modified as

$$W_i'' = |W_i|\,\frac{\sum\limits_{j=1}^{N}\left[W_j\,\delta_{\mathrm{K}}(B_j,B_i)\right]}{\sum\limits_{j=1}^{N}\left[|W_j|\,\delta_{\mathrm{K}}(B_j,B_i)\right]}\,. \tag{A.9}$$

The corresponding estimate $\widehat{F}_B''(b)$ would be the same as $\widehat{F}_B(b)$, while $\widehat{\sigma}_B''(b)$ would be lesser than $\widehat{\sigma}_B(b)$ (unless the original non-zero weights in bin $b$ all had the same sign).

These discrepancies are a symptom of the larger problem of mutual dependency between the modified datapoints. The reweightings in (A.8) and (A.9) do not change the variance of the estimate of $F_B(b)$; they simply move a non-zero component from the first additive term to the second in the right-hand-side of (A.2). As a special case, for the reweighting in (A.8), the weights of all the events in a given bin are perfectly correlated to each other post reweighting. The errors are underestimated by $\widehat{\sigma}_B'(b)$ and $\widehat{\sigma}_B''(b)$ because the formula in (A.7) does not capture the uncertainties in (A.4) stemming from the aforementioned mutual dependency.[42] This issue cannot be eliminated by a downsampling of the kind suggested in Ref. [35].

More generally, the standard statistical analysis apparatus in HEP cannot be assumed to automatically work with such non-IID datasets. For example, if the weight redistribution is not contained within individual histogram bins but spreads across bins, then $\widehat{F}_B(b)$ and $\widehat{F}_B(b')$, for $b \neq b'$, could be more strongly correlated than one would expect for IID datapoints.[43] In addition to the data-correlation issue raised here, (a) biases arising from smearing the weights over non-zero volumes in the phase space of $V$ and (b) ML training errors (where relevant) can also make it difficult to deploy the techniques discussed here.

**Can positive resampling techniques provide any statistical benefit?** In the binned example above, it is not obvious that there is any benefit to performing the redistribution of weights among the events, since the histogram heights stay the same as before (and consequently, so do the associated uncertainties). Removing the negative weights in a dataset, without an

---

[41]The uncertainty estimate getting reduced would not problematic if the original uncertainty estimate was sufficiently conservative, but that is not the case here.

[42]A roughly equivalent way to think about this is that one needs to account for the MC uncertainties in $W_i'$ and $W_i''$ in order to properly estimate the errors in $\hat{F}_B'(b)$ and $\hat{F}_B''(b)$.

[43]Such correlations have implications for the discussion around data "amplification" by generative models [62–65], and will be discussed in a future work [66].

associated reduction in uncertainties, does not necessarily constitute a solution to the negative weights "problem". Stated differently, the kind of local averaging of event-weights performed by a resampler is what happens in a histogram-based experimental analysis (where event-weights within each bin are aggregated) anyway. The MC uncertainties in such weight-averaging is the crux of the negative weights problem in the first place. So, it is not obvious that moving the weight-averaging step from the analysis stage into the simulation pipeline adds any value. However, it turns out that there could be a statistical benefit to such techniques, as explained below.

An important detail in this discussion is that the positive resampling and other related techniques all propose to reweight events at (or immediately after) the particle-level event generation stage. On the other hand, experimental analyses only have access to reconstruction-level simulated events. This has two implications. First, one can perform the weight redistribution using generator-level latent phase space information, and thereby have access to a bias–variance–correlation tradeoff [66] that is not available at the analysis stage. No claims are made here regarding whether this can be leveraged for a real-world statistical benefit.

Secondly, and arguably more importantly, it turns out that one can perform the following downsampling precedure for increasing the efficiency of a dataset: a) produce more events than needed, at the generator level, b) perform the weight-redistribution using positive resampling techniques, and c) downsample the resampled dataset using rejection reweighting (similar to suggestions in Refs. [34–36]), before the subsequent processing of the event, including detector simulation. The acceptance probability function in the downsampling step could simply be a small constant (independent of event-weight) for this procedure to be useful. The *oversample, reweight, then downsample* approach leads to reduced correlations between the accepted events, when compared to the *produce only what is needed and reweight* approach.[44] In this way one can achieve better uncertainties or sensitivities for a given number of events that undergo detector simulation. The mechanism and math behind this idea will be discussed in a future work, Ref. [66]. The greater the degree to which the dataset is downsampled, the greater the statistical benefit from this procedure, not counting the generator-level computational cost associated with the oversampling in step (a). Note that the standard error formulas for IID datasets will continue to underestimate the errors even after this downsapling procedure.[45]

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
