# Peer review of "ARCANE Reweighting: A Monte Carlo Technique to Tackle the Negative Weights Problem in Collider Event Generation"

_SciPost Physics_

## Round 1 · Referee Report · Anonymous (Referee 1) · 2025-5-15

Strengths

This paper provides a detailed description of a novel reweighting technique to address the pervasive problem of negative weights in the generation of Monte Carlo samples at higher orders in perturbation theory. It is concisely written and allows the reader to follow the logic of the formalism in great detail. Reference to a companion paper where a first example has been worked out helps a lot, especially when reading both papers next to each other.

Weaknesses

  1. In my opinion, the reference to a companion paper is less than optimal, as it leaves the reader with the feeling that the present paper is not self-contained. Given, however, the mathematical apparatus and the clarity in the presentation this is maybe appropriate. Having said that, I would have appreciated a simplistic one-dimensional example to allow the reader to develop some intuition about the method.
  2. While the companion paper features some examples, it is quite obvious that they are highly trivial. I wonder how the formalism scales with more complicated processes, where the phase space is more convoluted. I think it would be fair to state - maybe in the conclusions - that this needs to be studied.

Report

I think all criteria are met, and I would support publication as is - I leave it to the author to respond to my comments/suggestions at their convenience.

Recommendation

Publish (easily meets expectations and criteria for this Journal; among top 50%)

---

## Round 1 · Referee Report · Anonymous (Referee 2) · 2025-5-23

Strengths

  1. Very Relevant for HL-LHC simulations
  2. Nice and original idea
  3. Very nice summary section of the various method used to tackle the problem in the context of MC@NLO

Weaknesses

  1. Quite Long paper, especially compare to the actual amount of new idea/discussion present in the paper.

  2. The author decides to split his work into two papers, this one containing the idea and a second applying it in a simple (but practical) example. This makes this paper quite abstract and lacking of practical example.

  3. The targeted audience of the paper is not clear, the paper often switches between a focus on MC@NLO case and on a more general focus.

  4. The success of the method presented is not demonstrated in this paper. The author either refers to another paper or is using vague statement like "The success of Arcane (...) stems from the intuition that ...."

Report

The main strength, weakness are reported above.

On the science part, the paper is on solid ground, but the decision of splitting the paper from an actual implementation is questionable, even more since this paper can not be read without solid MC@NLO knowledge so it does not really address to a larger community.

Based on that, I would suggest the editor to ask the author to consider making a version of the paper which is not linked at all to MC@NLO and which can be non physics targeted.

Finally, here are a couple of minor points:

  • The paper complains abut the fact that POWHEG/MC@NLO-Delta are not generic enough, but it is hard to approve such statement (based on 5 years old reference) or at least one should contextualize that ARCANE is also not yet at such level.

  • The paper does not correctly define the weight normalization, this is actually important since equation (30) is actually not correct only for all normalization choice.

  • I would like to report a couple of typos: page 19 at point 2, this should be for all W different of zero, not only for the positive one page 17: procedues -> procedures

Recommendation

Accept in alternative Journal (see Report)

---

## Editorial Decision

awaiting_resubmission